# TT-SPARSE: Learning Sparse Rule Models with Differentiable Truth Tables

**Hans Farrell Soegeng** [1]   **Sarthak Ketanbhai Modi** [1]   **Thomas Peyrin** [1]

## Abstract

Interpretable machine learning is essential in high-stakes domains where decision-making requires accountability, transparency, and trust. While rule-based models offer global and exact interpretability, learning rule sets that simultaneously achieve high predictive performance and low, human-understandable complexity remains challenging. To address this, we introduce TT-SPARSE, a flexible neural building block that leverages differentiable truth tables as nodes to learn sparse, effective connections. A key contribution of our approach is a new soft TOPK operator with straight-through estimation for learning discrete, cardinality-constrained feature selection in an end-to-end differentiable manner. Crucially, the forward pass remains sparse, enabling each node (and the entire model) to be transformed exactly into compact, globally interpretable DNF/CNF Boolean formulas via Quine–McCluskey minimization. Extensive empirical results across 28 datasets spanning binary, multiclass, and regression tasks show that the learned sparse rules exhibit superior predictive performance with lower complexity compared to existing state-of-the-art methods. https://github.com/hansfarrell/tt-sparse

## 1. Introduction

In high-stakes deployments including healthcare, finance, public policy, and safety-critical engineering, interpretability is often a functional requirement, enabling accountability and auditability (Doshi-Velez & Kim, 2017; Seshia et al., 2022; Soegeng et al., 2025). In these regimes, post-hoc explainability methods that attempt to rationalize a black-box predictor can be brittle: the explanation may be misaligned with the true decision process (Slack et al., 2020; Adebayo et al., 2018). Rudin (2019) argues that, when possible, one

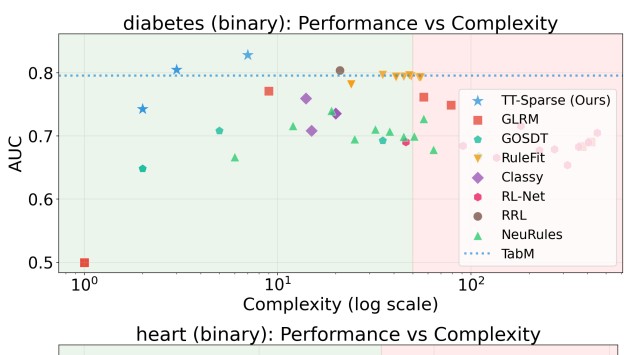

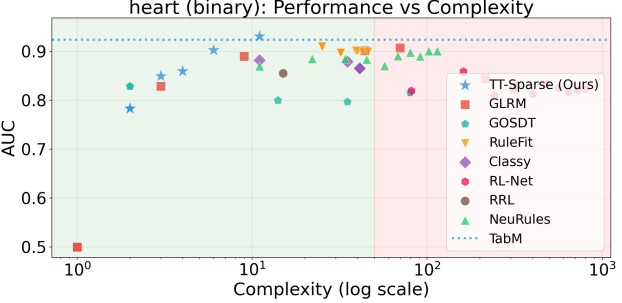

*Figure 1.* AUC score against rule log-complexity scatter plot of 8 interpretable models and TabM as SOTA black-box baseline across different hyperparameters, on diabetes and heart tabular datasets.

should prefer *inherently interpretable* models over explanations of opaque ones, where transparency is a property of the model itself.

Rule-based predictors are a natural target for such fundamental interpretability. However, interpretability is multifaceted. First, *global* interpretability means that a single set of rules governs the model's behavior for all inputs; this contrasts with *local* explanations that provide input-dependent rationales. Second, *exact* interpretability means that the extracted rules reproduce the model's inference exactly, rather than approximately (e.g., via feature-attribution scores). In this work, we focus on the demanding but practically valuable regime of **global and exact** interpretability, where the entire model's inference can be reduced to symbolic logic without approximation.

Crucially, interpretability does not guarantee human understandability. Human-subject evaluations demonstrate that cognitive load scales non-linearly; specifically, studies suggest that rule sets exceeding a complexity of 50 become

[1]School of Physical and Mathematical Sciences, Nanyang Technological University, Singapore. Correspondence to: Hans Farrell Soegeng <hans0048@e.ntu.edu.sg>.

*Proceedings of the 43rd International Conference on Machine Learning*, Seoul, South Korea. PMLR 306, 2026. Copyright 2026 by the author(s).

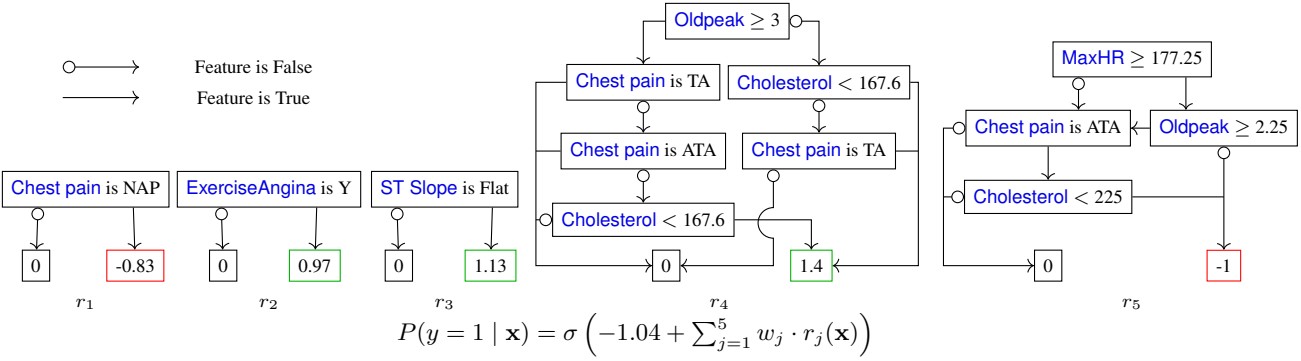

$$P(y = 1 \mid \mathbf{x}) = \sigma \left( -1.04 + \sum_{j=1}^{5} w_j \cdot r_j(\mathbf{x}) \right)$$

*Figure 2.* A TT-SPARSE model trained on Heart dataset converted to Boolean decision trees, achieving 91% test ROC-AUC score and complexity of 15. The sigmoid $\sigma(\cdot)$ function is applied to the final activated weights + intercept term to obtain the probability of heart disease existence between $[0, 1]$.

functionally unintelligible (Lage et al., 2019). Hence, our work focuses on producing rule sets that minimize *rule complexity while maintaining competitive predictive performance*. Figure 1 showcases this: our model TT-SPARSE achieves superior predictive performance while maintaining low rule complexity (well within the complexity threshold for human intelligibility of 50 indicated by the green region), outperforming other rule models and remaining competitive with the state-of-the-art deep tabular model, TabM (Gorishniy et al., 2025).

Despite their appeal, learning high-performing rule sets while minimizing complexity remains challenging. Classical approaches often rely on discrete, heuristic search (e.g., greedy rule induction or tree growth/pruning), which can become trapped in suboptimal solutions and may struggle to exploit modern hardware efficiently (Cohen, 1995; Breiman et al., 1984). More recent lines of work pursue global optimization objectives or neural-symbolic relaxations that enable gradient-based training, but they impose restrictive logical forms that limit expressivity (Petersen et al., 2022a; 2024).

Truth table-based nodes offer a direct route to exact symbolic conversion via standard minimization procedures such as Quine-McCluskey (Benamira et al., 2023; 2024). Crucially, truth table-based nodes provide high expressivity because they are capable of representing any Boolean function over their inputs, effectively acting as the discrete counterpart to standard neural network (NN) nodes. Unlike neural-symbolic relaxations that approximate discrete logic through continuous functions, truth table nodes function as fundamental building blocks that directly map input patterns to outputs. However, making such nodes trainable at scale requires solving a key bottleneck: selecting a sparse, meaningful subset of inputs for each node in a way that remains compatible with backpropagation.

We introduce **TT-SPARSE**, a differentiable rule-learning architecture built around *Learnable Truth Table (LTT)* nodes that can be transformed exactly into Boolean formulas (DNF/CNF) after training. Each LTT node learns a Boolean function over a small set of selected input features; the entire model is then a composition of these rule activations with a lightweight prediction head. The central technical challenge is that choosing which $k$ inputs feed each node is a discrete TOPK operation, which is non-differentiable. TT-SPARSE resolves this with a novel **soft TOPK** operator that provides a continuous relaxation while enforcing an exact cardinality constraint in expectation, enabling stable gradient flow through the connection-selection mechanism. Concretely, we use a straight-through strategy where the forward pass uses hard TOPK selections to preserve sparsity, while the backward pass uses the differentiable relaxation to update connection scores. After training, we enumerate each node's induced truth table and apply Quine-McCluskey (Quine, 1952; McCluskey, 1956) minimization to obtain compact symbolic rules, yielding global and exact interpretability.

Empirically, TT-SPARSE consistently achieves favorable performance-complexity trade-offs across 28 tabular benchmarks spanning binary, multiclass, and regression tasks, the only interpretable rule-learning model that natively supports all three within a single architecture while remaining competitive with the SOTA non-pretrained tabular model.

**Conflict of Interest Disclosure.** All authors are affiliated with Implicant (https://implicant.ai), a company that deploys TT-SPARSE for commercial applications. The model evaluated in this paper forms the core technology of that product.

**Contributions.** The main contributions of this work include: (1) We propose a novel, efficient, and fully **differentiable relaxation of the discrete TOPK operator**, enabling end-to-end gradient-based optimization of discrete feature routing via backpropagation with zero gradient variance. (2) We introduce the **TT-SPARSE layer**, a flexible neural building

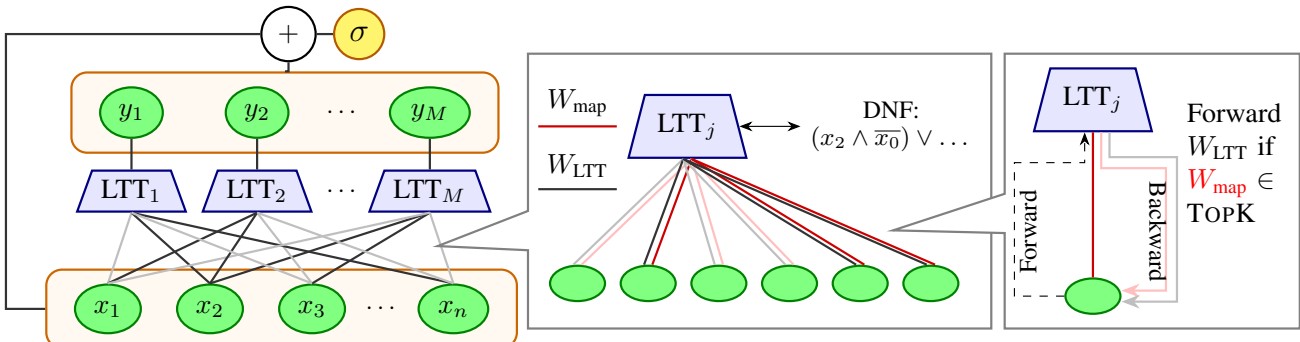

*Figure 3.* **Overview of the TT-SPARSE Architecture. (Left)** The hybrid model structure. The input vector is processed by a layer of Learnable Truth Table (LTT) nodes (blue trapezoids), which extract multi-feature Boolean rules. These outputs are concatenated with the raw input features to form the final prediction. **(Middle)** Each potential connection is parameterized by the logic weight $W_{\text{LTT}}$ and the mapping weight $W_{\text{map}}$. Darker lines highlight the "active" connections, those where $W_{\text{map}}$ belongs to the subset of the $k$-highest mapping weights for that node. Each LTT node is convertible to an equivalent CNF/DNF equation (See Figure 4). **(Right)** In the forward pass, the input is fed forward and multiplied by $W_{\text{LTT}}$ if the corresponding $W_{\text{map}}$ is part of TOPK. In the backward pass, gradients flow with the Soft TOPK relaxation, updating both $W_{\text{map}}$ and $W_{\text{LTT}}$. This design enables exact rule extraction while preserving gradient-based training through discrete connection selection.

block that leverages our Soft TOPK operator to dynamically learn sparse connections and extract multi-feature Boolean interactions. Each LTT node's learned Boolean function is converted exactly to DNF/CNF via truth table enumeration and Quine-McCluskey minimization, yielding global and exact interpretability with no post-hoc approximation (Figure 3). (3) Through extensive empirical validation on 28 datasets, we demonstrate that TT-SPARSE establishes a new **Pareto frontier in the performance-complexity landscape** while achieving predictive accuracy competitive with SOTA non-pretrained tabular model, TabM. Notably, TT-SPARSE is the only interpretable rule-learning model that natively supports all three task types (binary, multiclass classification, and regression) within a single architecture.

## 2. Related Work

In this section, we provide an overview of the literature on methods that achieve global and exact interpretability, as well as relaxations of the discrete TOPK function. We evaluate these approaches with TT-SPARSE in Section 4.

**Tree-Based Approaches.** While classical decision trees (Breiman et al., 1984) offer an intuitive approach, their greedy induction strategy often leads to suboptimal rule sets. To address this, GOSDT (Lin et al., 2020) employs dynamic programming to find provably optimal sparse trees, though at a high computational cost for large datasets. Conversely, ensemble methods like LightGBM (Ke et al., 2017) achieve state-of-the-art predictive performance but sacrifice interpretability by distributing the decision logic across hundreds of weaker trees. However, the tree topology inherently restricts each root-to-leaf path to a logical conjunction (AND) of splitting predicates.

**Heuristic-Guided Bottom-Up Rule Induction.** RIPPER (Cohen, 1995) is a classic technique that employs a greedy strategy to grow and prune rule sets, incrementally refining them to optimize predictive accuracy. Classy (Proença & van Leeuwen, 2020) advances this approach by utilizing the Minimum Description Length (MDL) principle to search for compact probabilistic rule lists without the need for hyperparameter tuning. Other approaches, such as GLRM (Wei et al., 2019) and RuleFit (Friedman & Popescu, 2008) improve scalability by leveraging column generation or tree ensembles to identify candidate rules. However, similar to the limitation with trees, these methods remain constrained by discrete heuristic generation: they build rules by iteratively adding conjunctions (ANDs), lacking the native flexibility to learn complex, nested Boolean structures (e.g., ORs) directly within a single rule unit.

**Neuro-Symbolic Approaches.** Neuro-symbolic models aim to combine the generalization power of neural networks with symbolic interpretability. DILP (Evans & Grefenstette, 2018) and RRL (Wang et al., 2021) learn a set of discrete rules by projecting logical structures into a differentiable space and applying gradient-based optimization. RL-Net (Dierckx et al., 2023) uses a neural architecture that mimics rule lists by enforcing a hierarchical activation structure. Most recently, NeuRules (Xu et al., 2025) integrate feature discretization, rule construction, and rule ordering into a single end-to-end differentiable pipeline.

**Differentiable TOPK.** Prior differentiable TopK approaches rely on iterative Optimal Transport solvers with the Sinkhorn algorithm (Xie et al., 2020; Cuturi, 2013) or stochastic noise injection (Cordonnier et al., 2021; Berthet et al., 2020), which introduces significant computational overhead or gradient variance. Most recently, Petersen et al.

(2022b) used these novel differentiable TOPK operators to define a top-$k$ cross-entropy loss, enabling the direct end-to-end optimization of top-$k$ classification accuracy, achieving state-of-the-art fine-tuning results on ImageNet.

## 3. TT-SPARSE

In this section, we introduce TT-SPARSE, a **T**ruth **T**able-based **Sparse** neural block. Each node in the TT-SPARSE layer can be understood as a learnable truth table logic over its inputs; we call them Learnable Truth Table (LTT) nodes interchangeably. This block is the core interpretable unit that transforms input features into Boolean rules. Let $\vec{x}$ be the $n$-dimensional input vector to the layer, and there are $M$ LTT nodes in the layer.

The TT-SPARSE layer comprises of 3 components: the connection selection, the linear operation of the selected inputs, and the binarization. The connection selection mechanism leverages the soft TOPK operator we introduce to enable learning by backpropagation of the loss by relaxing the discrete, non-differentiable selection of the traditional TOPK.

### 3.1. Components

#### 3.1.1. SOFT TOPK

We define the operator

$$S_k : \mathbb{R}^n \to \mathcal{P}_k \subset [0,1]^n$$

where $\vec{s} \in \mathbb{R}^n$ denotes an input score vector (in practice, a column of the connection matrix $W_{\mathrm{map}}$; see Section 3.1.2) and $\mathcal{P}_k = \{\vec{\pi} \in [0,1]^n : \sum_{i=1}^n \pi_i = k\}$.

The hard TOPK operator can be viewed as finding the vector $\vec{\pi}$ within the feasible set that maximizes $\vec{s}^\top \vec{\pi}$, achieved by assigning the components of $\vec{\pi}$ corresponding to the $k$ largest values of $\vec{s}$ to 1, while setting the rest to 0. Since this discrete selection is not differentiable, we introduce an entropic regularizer to find a selection probability vector $\vec{\pi} \in [0,1]^n$ that is closest to the scores $\vec{s}$ (maximizing the dot product) while maintaining smoothness and summing exactly to $k$. The entropic regularizer is weighted by a predetermined parameter temperature $\tau > 0$. This leads us to the optimization objective

$$\vec{\pi}^* = \underset{\vec{\pi} \in [0,1]^n}{\mathrm{argmax}} \left[ \sum_{i=1}^n \pi_i s_i + \tau \sum_{i=1}^n H(\pi_i) \right] \quad \text{s.t.} \quad \sum_{i=1}^n \pi_i = k \tag{1}$$

where $H(\pi_i) = -\pi_i \log \pi_i - (1 - \pi_i) \log(1 - \pi_i)$ is the binary entropy. Solving via Lagrange multipliers (Appendix C), the solution $S_k(\vec{s}) := \vec{\pi}^*$ has components

$$\pi_i^* = \sigma\left( \frac{s_i}{\tau} + c \right) \tag{2}$$

where $\sigma(\cdot)$ is the sigmoid and $c \in \mathbb{R}$ is the unique root of $f(c) = \sum_{j=1}^n \sigma(\frac{s_j}{\tau} + c) - k = 0$, found by bisection.

This allows exact gradients to be derived by implicit differentiation. Let $d_i = \sigma'(\frac{s_i}{\tau} + c) = \pi_i^*(1 - \pi_i^*)$ and $\vec{d} = [d_1, \ldots, d_n]^\top$. Then the element-wise partials are:

$$\frac{\partial \pi_i}{\partial s_j} = \frac{1}{\tau} d_i \left( \delta_{ij} - \frac{d_j}{\|\vec{d}\|_1} \right) \tag{3}$$

$$\frac{\partial \pi_i}{\partial k} = \frac{d_i}{\|\vec{d}\|_1} \tag{4}$$

which yield the full Jacobian and gradient in matrix form:

$$\nabla_{\vec{s}} S_k(\vec{s}) = \frac{1}{\tau} \left[ \mathrm{diag}(\vec{d}) - \frac{\vec{d}\,\vec{d}^\top}{\|\vec{d}\|_1} \right] \tag{5}$$

$$\nabla_k S_k(\vec{s}) = \frac{\vec{d}}{\|\vec{d}\|_1} \tag{6}$$

The complete derivation of the partials can be found in Section C.2. The gradient is scaled by $\frac{1}{\tau}$, similar to the role of temperature in softmax. As $\tau \to 0$, the relaxation approaches the hard TOPK; the effect of $\tau$ on the selection weights is illustrated in Figure 6. Implementation details are in Section C.4.

The closed-form sigmoid solution is critical for TT-SPARSE: since the operator is applied independently to each of the $M$ LTT nodes at every training step, both memory footprint and gradient stability compound across the layer. Table 1 compares against the Sinkhorn-based optimal transport relaxation (Xie et al., 2020) and the perturbed maximizer (Cordonnier et al., 2021). While Sinkhorn is also deterministic (zero gradient variance), its iterative solver incurs $10\times$ higher latency and $316\times$ more memory, which is prohibitive when applied $M$ times per forward pass. The Perturbed operator matches our throughput but requires stochastic noise injection, introducing substantial gradient variance that destabilizes connection routing over long training runs, and consuming $73\times$ more memory to store the Monte Carlo samples.

*Table 1.* Differentiable TOPK operator comparison (GPU, `float32`, batch 64, mean across 10 configurations). Perturbed gradient quality reported for 32 / 256 MC samples.

|  | Ours | Perturbed | Sinkhorn |
|---|---|---|---|
| Forward (ms) | 4.08 | 3.46 | 41.0 |
| Backward (ms) | 1.25 | 1.15 | 32.2 |
| Memory (MB) | 3.4 | 247.4 | 1,075.7 |
| Throughput (s/s) | 36,177 | 31,939 | 1,980 |
| Grad. Variance | 0.0 | 200.3 / 22.1 | 0.0 |
| Loss Std. | 0.0 | 1.80 / 0.39 | 0.0 |

### 3.1.2. LTT NODES

Let $\vec{x} \in \mathbb{R}^n$ denote the input to the TT-SPARSE layer with $M$ LTT nodes. The block has 2 learnable modules: the connection mapping represented by a matrix $W_{\text{map}} \in \mathbb{R}^{n \times M}$ and the truth table logic represented by matrix $W_{\text{LTT}} \in \mathbb{R}^{n \times M}$ (augmented by a bias vector $b \in \mathbb{R}^M$).

For each LTT node $j \in \{1, \ldots, M\}$, the objective is to select a sparse subset of $k$ input features to form a truth table of size $2^k$ and learn effective truth table logic over the features. To determine the active inputs for node $j$, we define a selection mask vector $m^{(j)} \in \{0, 1\}^n$.

In the forward pass, this mask is discrete. It selects the indices corresponding to the top-$k$ values in the $j$-th column of the mapping matrix:

$$m_{\text{forward},i}^{(j)} = \begin{cases} 1 & \text{if } W_{\text{map}}[i,j] \in \text{TopK}(W_{\text{map}}[:,j]) \\ 0 & \text{otherwise} \end{cases}$$

for $i \in [1, \ldots, n]$.

The intermediary output of each LTT node is the linear combination of the active features. Let $\mathcal{I}_j = \{i : m_i^{(j)} = 1\}$ with $|\mathcal{I}_j| = k$ denote the selected index set for node $j$. Then:

$$z_j(\vec{x}) = \sum_{i \in \mathcal{I}_j} W_{\text{LTT}}[i,j] \cdot x_i + b_j \tag{7}$$

However, the traditional TOPK operation is discrete and not differentiable, so without relaxation, the connection matrix will not update to find better connections for the LTT logic. Thus, we leverage the soft TOPK operator $S_k$ we propose in 3.1.1 to replace $m^{(j)}$ in the backward pass with $m_{\text{backward}}^{(j)} = S_k(W_{\text{map}}[\cdot, j])$, where the score vector $\vec{s} = W_{\text{map}}[\cdot, j]$.

Consequently, the partial derivatives w.r.t. the connection weights are computed via the chain rule. The gradient flow to the connection matrix is given by

$$\frac{\partial \mathcal{L}}{\partial W_{\text{map}}[\cdot, j]} = \frac{\partial \mathcal{L}}{\partial z_j} \cdot \frac{\partial z_j}{\partial m_{\text{backward}}^{(j)}} \cdot \frac{\partial S_k(W_{\text{map}}[\cdot, j])}{\partial W_{\text{map}}[\cdot, j]} \tag{8}$$

The first term $\frac{\partial \mathcal{L}}{\partial z_j}$ is the upstream loss gradient, representing how much the overall loss changes w.r.t. the node's pre-binarization output. The second term $\frac{\partial z_j}{\partial m_{\text{backward}}^{(j)}} = \vec{x} \odot W_{\text{LTT}}[\cdot, j]$ is an $n$-dimensional vector representing the potential contribution of each feature if selected. The third term $\frac{\partial S_k(W_{\text{map}}[\cdot, j])}{\partial W_{\text{map}}[\cdot, j]}$ is the Jacobian of the Soft TOPK operator, $J_{S_k} \in \mathbb{R}^{n \times n}$ given by Equation (5) (with $\vec{s} = W_{\text{map}}[\cdot, j]$). Similar to the connection weights, the gradient for the truth table weights $W_{\text{LTT}}[i, j]$ is scaled by the selection probability $\pi_i^{(j)} = [S_k(W_{\text{map}}[\cdot, j])]_i$, so features with higher selection mass receive stronger weight updates. Thus, both $W_{\text{map}}$

and $W_{\text{LTT}}$ are jointly optimized through the soft TOPK relaxation: the former learns *which* features to route, while the latter learns *how* to combine them. The full derivation is provided in Appendix D. This mechanism allows the TT-SPARSE layer to dynamically re-route connections into the LTT node while learning the LTT logic during training.

Finally, to function as a Boolean logic gate, the continuous output $z_j$ is binarized to $y_j = \mathbf{1}_{(z_j > 0)}$. Differentiability is maintained with a straight-through estimator (Bengio et al., 2013).

## 3.2. TT-SPARSE Design

The architecture of the overall model is designed as a hybrid neural block that integrates multi-feature interaction logic through the TT-SPARSE layer with linear single-feature.

The layer processes an $n$-dimensional input vector $\vec{x}$ to produce $M$ binary rule activations $\vec{y} \in \{0, 1\}^M$ via the LTT nodes. Then, we implement a skip-concatenation, so the final representation $\vec{h}$ is formed by concatenating the raw input features with the LTT rule outputs: $\vec{h} = [\vec{x} \parallel \vec{y}] \in \mathbb{R}^{n+M}$. This combined vector is then fed into a final classifier or regressor layer $f_{\text{cls}}(\vec{h}) = \sigma(W_{\text{cls}}\vec{h} + b_{\text{cls}})$, depending on the task (binary/multiclass/regression). Because $f_{\text{cls}}$ is linear over the concatenation, each weight corresponds to either a multi-feature Boolean rule (from $\vec{y}$, extracted via truth table enumeration) or a single-feature (unary) rule (from $\vec{x}$, readable directly as a weighted threshold). The extracted rule set includes both terms with exact weights, preserving full interpretability.

While the TT-SPARSE layer selects several inputs per node, not all connections or all nodes may be essential for a given task. To identify the most compact and effective rule set, we implement a post-hoc iterative magnitude pruning after the initial training phase. Connections in the TT-SPARSE layer are removed based on the weights' $L_1$ norm, followed by an iterative fine-tuning of the non-zeroed weights. Throughout this refinement process, the selected connections of the LTT nodes remain fixed to the selected indices by $W_{\text{map}}$ during training. This refinement phase focuses on optimizing the weights of the established logical structure, finding the lightest and most effective rule logic for the given task.

## 3.3. Rule Extraction

Each node $j$ computes a learnable Boolean function over its $k$ selected inputs. Let $\mathcal{I}_j = \{i_1, \ldots, i_k\} \subset \{1, \ldots, n\}$ denote the indices selected by the top-$k$ operator on $W_{\text{map}}$. To extract the explicit Boolean formula, we enumerate all $2^k$ binary assignments to these $k$ inputs. For a binary vector $v \in \{0, 1\}^k$, component $v_\ell$ corresponds to the input feature $\vec{x}$ at index $i_\ell \in \mathcal{I}_j$. Let $w^{(j)} \in \mathbb{R}^k$ be the subvector of

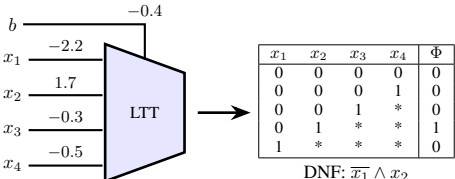

*Figure 4.* Conversion of an LTT node to DNF by truth table enumeration of $2^n$ input combinations and obtaining the binary outputs with the LTT weight and bias parameters.

$W_{\text{LTT}}[:, j]$ restricted to $\mathcal{I}_j$. The localized evaluation is:

$$y_j(v) = \mathbf{1}\left(v^\top w^{(j)} + b_j > 0\right) = \mathbf{1}\left(z_j > 0\right) \qquad (9)$$

The truth table $\mathcal{T}_j = \{(v^{(i)}, y_j(v^{(i)}))\}_{i=1}^{2^k}$ records the output for every assignment, where $v^{(1)}, \ldots, v^{(2^k)}$ enumerates all $2^k$ binary vectors in $\{0, 1\}^k$ (e.g., $000, 001, 010, \ldots$). The subset $\mathcal{M}_j = \{v^{(i)} \mid y_j(v^{(i)}) = 1\}$ are the *minterms* of the Boolean function.

To minimize rule complexity, we identify *Don't-Care Terms* (DCTs): input combinations ($v^{(i)}$) that do not appear in the training data. Following the identification of minterms and DCTs, the truth table is synthesized into a minimal DNF expression via the Quine-McCluskey (QMC) algorithm (Quine, 1952; McCluskey, 1956) (Algorithm 1). By optimally assigning binary values to these DCTs, the algorithm minimizes the number of implicants and literals in the resulting rule while maintaining exact fidelity on the observed training distribution.

After QMC minimization, each node's Boolean function is expressed in Disjunctive Normal Form (DNF): a disjunction (OR) of *implicants*, where each implicant is a conjunction (AND) of *literals* (Boolean conditions on individual features). We define the **complexity** of a rule set as the total number of literals across all rules in the model, including unary rules and the bias term. For multiclass tasks, a rule contributes to the count exactly once if it holds a non-zero weight for any class.

**Per-node complexity.** Each node with fan-in $k$ computes a linearly separable (threshold) Boolean function of its $k$ inputs. By Sperner's theorem, the maximum number of prime implicants of any such function is $\binom{k}{\lfloor k/2 \rfloor}$ (Muroga, 1971). Since a node's total literal count in its minimized DNF directly equals its contribution to overall model complexity, Table 2 characterizes learned per-node complexity by extracting and minimizing every non-trivial node's Boolean function via QMC across all 28 datasets and 5 seeds. The "with DCT" column marks input patterns unobserved in training data as don't-cares, as the full extraction pipeline.

Table 2 shows that mean per-node literal count remains

*Table 2.* Mean per-node rule complexity after QMC minimization, aggregated across 28 datasets and 5 seeds. Bound: worst-case implicants for threshold functions (Muroga, 1971). DCT: don't-care terms from unobserved training patterns.

| $k$ | Bound | Without DCT | | With DCT | |
|---|---|---|---|---|---|
| | | Implicants | Literals | Implicants | Literals |
| 2 | 2 | 1.20 | 1.73 | 1.14 | 1.53 |
| 3 | 3 | 1.59 | 3.02 | 1.38 | 2.25 |
| 4 | 6 | 2.33 | 5.57 | 1.74 | 3.40 |
| 5 | 10 | 3.63 | 10.51 | 2.22 | 5.06 |
| 6 | 20 | 5.65 | 19.24 | 2.77 | 7.13 |

well below the tight worst-case bound at every fan-in. At $k=6$, nodes average 5.7 implicants (19.2 literals) versus a bound of 20 implicants, and don't-care optimization further reduces this to 2.8 implicants (7.1 literals). This confirms that individual nodes converge to functions that use only a fraction of their representational capacity, and that the training-data-aware DCT simplification yields rules roughly half as complex while providing regularization. The total model complexity (reported in Tables 3–5) is then the sum of per-node literals for all rules.

By modeling the node truth table logic as a thresholded linear combination of the selected inputs, TT-SPARSE requires only $n + 1$ learnable parameters (one weight per selected input plus a bias term) to represent the logic, a linear complexity compared to DiffLogicNet (Petersen et al., 2022a) double-exponential and DWN (Bacellar et al., 2024) exponential complexities.

DiffLogicNet introduces a differentiable relaxation of the Boolean logic to implement binary gates as nodes, by learning a probability distribution over the set of all possible binary operators for a given number of inputs. While effective for 2-input gates, this approach scales poorly as the number of possible Boolean functions grows double exponentially with the number of inputs, $(2^{2^n})$.

The DWN model (Bacellar et al., 2024) addresses this scalability bottleneck by parameterizing nodes as Lookup Tables (LUTs) with $2^n$ parameters and Extended Finite Differences for gradient estimation. However, LUTs are position-dependent: permuting selected features invalidates the learned weights, causing inefficiency during connection search. In contrast, the TT-SPARSE linear combination is commutative, making nodes robust to input reordering. Additionally, pruning a LUT input requires constraining half the table via Shannon expansion, whereas TT-SPARSE simply pushes a weight to zero via $L_1$. We validate these advantages empirically in Tables 3–5.

*Table 3.* Binary Classification Results: AUC ↑ (top) and Complexity ↓ (bottom) for each dataset. Standard error in subscript. Best values among interpretable models in **bold**. The last row reports the avg. rank ↓ of each model's AUC across all datasets, as calculated for the Friedman statistical test. *OOM* indicates out-of-memory during training.

| Dataset | TT-SPARSE | GOSDT | GLRM | RuleFit | Classy | NeuRules | RL-Net | RRL | DWN | TabM |
|---|---|---|---|---|---|---|---|---|---|---|
| bank | $\mathbf{.9096}_{00}$ | $.6662_{01}$ | $.8928_{00}$ | $.8963_{01}$ | $.9042_{00}$ | $.8648_{01}$ | $.6917_{01}$ | $.8913_{01}$ | $.8480_{01}$ | $.9351_{00}$ |
| | $112_{37}$ | $\mathbf{5}_{0}$ | $38_{2}$ | $58_{20}$ | $370_{13}$ | $902_{45}$ | $1318_{6}$ | $73_{7}$ | $533_{31}$ | - |
| blood | $\mathbf{.7554}_{02}$ | $.6155_{04}$ | $.7093_{05}$ | $.7059_{05}$ | $.6628_{03}$ | $.7116_{02}$ | $.5618_{05}$ | $.6712_{04}$ | $.6545_{05}$ | $.7193_{04}$ |
| | $15_{8}$ | $15_{5}$ | $30_{5}$ | $43_{7}$ | $9_{2}$ | $165_{3}$ | $201_{0}$ | $\mathbf{5}_{2}$ | $465_{29}$ | - |
| calhousing | $\mathbf{.9307}_{00}$ | $.8111_{01}$ | $.9205_{01}$ | $.9259_{00}$ | $.9072_{01}$ | $.8804_{01}$ | $.8004_{03}$ | $.8684_{01}$ | $.7961_{01}$ | $.9667_{00}$ |
| | $62_{11}$ | $\mathbf{25}_{0}$ | $108_{7}$ | $131_{46}$ | $457_{17}$ | $229_{8}$ | $88_{1}$ | $45_{3}$ | $572_{21}$ | - |
| compas | $\mathbf{.7301}_{01}$ | $.6765_{01}$ | $.7236_{01}$ | $.7270_{01}$ | $.7080_{01}$ | $.7129_{01}$ | $.6851_{02}$ | $.6825_{01}$ | $.6629_{01}$ | $.7311_{01}$ |
| | $\mathbf{9}_{2}$ | $13_{0}$ | $38_{4}$ | $31_{16}$ | $39_{2}$ | $280_{6}$ | $961_{1}$ | $17_{5}$ | $1358_{28}$ | - |
| covertype | $.8554_{00}$ | *OOM* | $.8437_{00}$ | $.8465_{00}$ | $\mathbf{.8913}_{00}$ | $.8320_{00}$ | $.7672_{01}$ | $.8522_{00}$ | $.8002_{00}$ | $.9801_{00}$ |
| | $194_{31}$ | | $49_{1}$ | $\mathbf{39}_{6}$ | $2183_{102}$ | $284_{28}$ | $194_{16}$ | $325_{8}$ | $464_{16}$ | - |
| cc_default | $\mathbf{.7915}_{01}$ | $.7011_{01}$ | $.7705_{01}$ | $.7802_{00}$ | $.7631_{01}$ | $.7596_{01}$ | $.7233_{00}$ | $.7454_{02}$ | $.6601_{02}$ | $.7883_{01}$ |
| | $128_{19}$ | $\mathbf{6}_{2}$ | $56_{24}$ | $110_{41}$ | $116_{3}$ | $281_{20}$ | $440_{2}$ | $100_{20}$ | $445_{18}$ | - |
| creditg | $.7919_{04}$ | $.5788_{02}$ | $.6851_{04}$ | $\mathbf{.7948}_{04}$ | $.6768_{02}$ | $.7313_{05}$ | $.6625_{04}$ | $.6745_{05}$ | $.7234_{07}$ | $.7329_{03}$ |
| | $54_{36}$ | $27_{6}$ | $56_{5}$ | $97_{24}$ | $\mathbf{13}_{2}$ | $781_{148}$ | $246_{0}$ | $35_{8}$ | $2444_{36}$ | - |
| diabetes | $\mathbf{.8208}_{02}$ | $.6443_{03}$ | $.7796_{02}$ | $.8024_{01}$ | $.7260_{02}$ | $.6854_{05}$ | $.6442_{02}$ | $.7609_{03}$ | $.7186_{04}$ | $.7957_{01}$ |
| | $14_{9}$ | $\mathbf{5}_{2}$ | $87_{10}$ | $19_{7}$ | $14_{3}$ | $18_{1}$ | $91_{0}$ | $24_{7}$ | $588_{26}$ | - |
| electricity | $.8823_{00}$ | $.7734_{00}$ | $.8705_{00}$ | $.8745_{00}$ | $\mathbf{.8895}_{00}$ | $.8397_{01}$ | $.7459_{01}$ | $.8387_{01}$ | $.8160_{00}$ | $.9666_{00}$ |
| | $99_{29}$ | $\mathbf{20}_{0}$ | $61_{3}$ | $186_{32}$ | $731_{12}$ | $240_{6}$ | $1121_{0}$ | $56_{6}$ | $2913_{96}$ | - |
| eye | $\mathbf{.6312}_{02}$ | $.5836_{00}$ | $.6278_{01}$ | $.6216_{01}$ | $.5913_{02}$ | $.5981_{01}$ | $.5815_{01}$ | $.5930_{01}$ | $.5711_{02}$ | $.6799_{02}$ |
| | $76_{43}$ | $\mathbf{8}_{2}$ | $16_{2}$ | $46_{18}$ | $38_{6}$ | $715_{24}$ | $1921_{0}$ | $64_{11}$ | $340_{19}$ | - |
| heart | $\mathbf{.9308}_{01}$ | $.8140_{01}$ | $.9149_{02}$ | $.9151_{01}$ | $.8749_{01}$ | $.8663_{01}$ | $.8301_{01}$ | $.8661_{03}$ | $.8719_{02}$ | $.9236_{02}$ |
| | $11_{15}$ | $\mathbf{2}_{0}$ | $39_{6}$ | $34_{16}$ | $31_{2}$ | $100_{9}$ | $641_{0}$ | $29_{10}$ | $474_{20}$ | - |
| income | $\mathbf{.9147}_{00}$ | $.7341_{01}$ | $.8958_{00}$ | $.9121_{00}$ | $.8799_{00}$ | $.8800_{01}$ | $.7707_{02}$ | $.8962_{01}$ | $.8642_{01}$ | $.9230_{00}$ |
| | $152_{40}$ | $\mathbf{11}_{2}$ | $28_{2}$ | $155_{47}$ | $684_{22}$ | $413_{58}$ | $4345_{1351}$ | $307_{18}$ | $1045_{23}$ | - |
| jungle | $.8917_{00}$ | $.7804_{01}$ | $.8904_{00}$ | $.8630_{01}$ | $\mathbf{.9476}_{00}$ | $.8702_{01}$ | $.7685_{03}$ | $.8781_{01}$ | $.7900_{01}$ | $.9970_{00}$ |
| | $262_{75}$ | $\mathbf{27}_{4}$ | $51_{4}$ | $103_{48}$ | $891_{25}$ | $163_{1}$ | $421_{0}$ | $28_{4}$ | $440_{25}$ | - |
| road_safety | $.7764_{00}$ | $.7197_{01}$ | $.7603_{00}$ | OOM | $\mathbf{.8435}_{00}$ | $.7842_{01}$ | $.7465_{01}$ | $.8293_{01}$ | $.7295_{02}$ | $.8852_{01}$ |
| | $236_{54}$ | $15_{8}$ | $24_{1}$ | | $1348_{33}$ | $186_{9}$ | $1948_{38}$ | $119_{10}$ | $425_{13}$ | - |
| Avg. rank ↓ | **1.53** | 8.27 | 3.6 | 3.07 | 3.8 | 4.8 | 7.93 | 4.87 | 7.13 | |

## 4. Experiments

We validate TT-SPARSE on 28 tabular datasets (14 binary, 7 multiclass, 7 regression), evaluating whether it achieves Pareto-competitive performance-complexity trade-offs against interpretable baselines and how it compares to TABM (Gorishniy et al., 2025), the SOTA non-pretrained tabular model.

**Datasets and Metrics.** We evaluate ROC-AUC for classification (One-vs-Rest for multiclass) to judge predictive performance while handling class imbalance and $R^2$ for regression. All models are trained with cross-entropy loss for classification and MSE for regression.

**Baselines.** We compare against **GOSDT** (Lin et al., 2020), **GLRM** (Wei et al., 2019), **RuleFit** (Friedman & Popescu, 2008), **Classy** (Proença & van Leeuwen, 2020), **NeuRules** (Xu et al., 2025), **RL-Net** (Dierckx et al., 2023), **RRL** (Wang et al., 2021), and **DWN** (Bacellar et al., 2024). We use the rule-set complexity metric defined in Section 3.3.

We visualize full performance-complexity Pareto frontiers in Appendix E; Tables 3–5 report the best Pareto-optimal configuration per model. Due to architectural limitations, GLRM and RuleFit cannot handle multiclass, and the remaining models are limited to classification.

For all binary classification datasets except covertype, jungle, and road_safety (Table 3), TT-SPARSE consistently pushes the Pareto limit, achieving comparable or better performance with lower complexity. While GOSDT often yields the lowest complexity rules with sparse decision trees, it frequently suffers from performance issues. For some datasets (blood, compas, cc_default, creditg, diabetes, heart, income), TT-SPARSE remains competitive with the SOTA DL baseline TabM, notably surpassing it on the Heart dataset with low complexity.

Multiclass classification (Table 4) highlights the strongest advantage of our approach. With GLRM and RuleFit unable to operate in this setting, TT-SPARSE comfortably outperforms the other models in all datasets as the Pareto

*Table 4.* Multiclass Classification Results: AUC ↑ (top) and Complexity ↓ (bottom) for each dataset. Standard error in subscript. Best values among interpretable models in **bold**. The last row reports the avg. rank ↓ of each model's AUC across all datasets, as calculated for the Friedman statistical test.

| Dataset | TT-Sparse | GOSDT | Classy | NeuRules | RL-Net | DWN | TabM |
|---|---|---|---|---|---|---|---|
| car | **.9901**$_{.00}$ $252_{110}$ | .6636$_{.03}$ **41**$_{12}$ | .9727$_{.01}$ $136_6$ | .8930$_{.02}$ $227_{11}$ | .8631$_{.03}$ $73_6$ | .6967$_{.06}$ $2445_{63}$ | 1.0000$_{.00}$ - |
| ecoli | **.9663**$_{.01}$ $148_{77}$ | .8315$_{.05}$ **16**$_4$ | .8968$_{.02}$ $34_4$ | .9340$_{.01}$ $201_3$ | .7551$_{.05}$ $36_0$ | .8841$_{.02}$ $1358_3$ | .9629$_{.01}$ - |
| iris | **.9993**$_{.00}$ $26_{17}$ | .9500$_{.02}$ **7**$_2$ | .9633$_{.02}$ $12_0$ | .9883$_{.01}$ $25_1$ | .9040$_{.05}$ $301_0$ | .8090$_{.04}$ $2406_{68}$ | .9963$_{.01}$ - |
| penguins | **1.0000**$_{.00}$ **6**$_1$ | .9519$_{.03}$ $15_2$ | .9795$_{.02}$ $19_3$ | .9893$_{.01}$ $212_8$ | .9551$_{.02}$ $50_2$ | .9503$_{.02}$ $1065_{55}$ | .9998$_{.00}$ - |
| satimage | **.9802**$_{.00}$ $157_{18}$ | .8408$_{.01}$ **17**$_3$ | .9587$_{.00}$ $408_{12}$ | .9488$_{.01}$ $378_9$ | .8094$_{.03}$ $740_2$ | .8652$_{.02}$ $554_{26}$ | .9925$_{.00}$ - |
| wine | **1.0000**$_{.00}$ **7**$_2$ | .9592$_{.01}$ $16_3$ | .9430$_{.03}$ $13_0$ | .9807$_{.02}$ $95_6$ | .9543$_{.06}$ $1121_0$ | .9528$_{.01}$ $795_{21}$ | 1.0000$_{.00}$ - |
| yeast | **.8457**$_{.02}$ $77_{14}$ | .6767$_{.02}$ **37**$_6$ | .7809$_{.02}$ $135_{11}$ | .7787$_{.01}$ $307_8$ | .7082$_{.03}$ $42_5$ | .6721$_{.04}$ $3050_{41}$ | .8564$_{.02}$ - |
| Avg. rank ↓ | **1** | 4.71 | 3 | 2.43 | 4.71 | 5.14 | |

*Table 5.* Regression Results: $R^2$ ↑ (top) and Complexity ↓ (bottom) for each dataset. Standard error in subscript. Best values among interpretable models in **bold**. The last row reports the avg. rank ↓ of each model's $R^2$ across all datasets, as calculated for the Friedman statistical test.

| Dataset | TT-Sparse | GLRM | RuleFit | TabM |
|---|---|---|---|---|
| abalone | **.5328**$_{.01}$ $43_{16}$ | .4103$_{.02}$ $53_6$ | .5324$_{.01}$ **23**$_7$ | .5494$_{.02}$ - |
| bike | .6956$_{.01}$ $299_{21}$ | .1879$_{.05}$ **50**$_5$ | **.7380**$_{.01}$ $127_{27}$ | .9547$_{.00}$ - |
| boston | .8566$_{.01}$ $77_{41}$ | .6541$_{.10}$ **74**$_{14}$ | **.8576**$_{.05}$ $89_{17}$ | .9042$_{.01}$ - |
| california | **.7102**$_{.01}$ $135_{30}$ | .5473$_{.04}$ $197_{20}$ | .7006$_{.01}$ **101**$_{34}$ | .8432$_{.01}$ - |
| crime | .3647$_{.08}$ $3003_{22}$ | .4893$_{.04}$ **68**$_5$ | **.6540**$_{.01}$ $79_{33}$ | .6513$_{.02}$ - |
| parkinsons | .9232$_{.01}$ $103_{90}$ | .8598$_{.02}$ **96**$_6$ | **.9270**$_{.01}$ $82_{15}$ | .9909$_{.00}$ - |
| wine_reg | **.3121**$_{.02}$ $111_{46}$ | .2151$_{.04}$ **59**$_8$ | .3090$_{.03}$ $85_{21}$ | .3629$_{.10}$ - |
| Avg. rank ↓ | 1.71 | 2.86 | **1.43** | |

a dedicated learnable mapping module $W_{\mathrm{map}}$ to determine which $k$ inputs each node receives, rather than selecting connections by weight magnitude alone. Third, within that mapping, it employs a differentiable Soft TopK operator to select the $k$ most important distinct inputs, rather than a slot-based Softmax as in DWN (Bacellar et al., 2024). The first two choices are validated here; the third is evaluated separately below.

We compare against three baselines that share the same LTT architecture and QMC extraction pipeline, differing only in how connectivity is determined. **FC + $L_0$ gates** (Louizos et al., 2018) learns stochastic binary gates with an $L_0$ penalty on gate openness, applying sparsification during training but without a fixed fan-in guarantee. **FC + $L_1$ + prune** applies $L_1$ regularization during training and then iteratively prunes by magnitude until per-node fan-in permits extraction. Both approaches attempt to reach the required fan-in through generic sparsification rather than enforcing it structurally. **Magnitude TopK** enforces the same fan-in $k$ as TT-Sparse, but selects the $k$ highest-magnitude LTT weights per node after $L_1$-regularized training, without a separate learned mapping. We sweep $\lambda \in \{0, \ldots, 10\}$ across 5 seeds on 4 datasets spanning all three task types (Table 6).

frontier, and is very close to achieving TabM performance while maintaining full transparency. TT-Sparse also shows strong competitiveness for the regression tasks (Table 5), effectively outperforming GLRM and underperforming Rule-Fit in rank by a thin margin.

### 4.1. Ablation Study

**TopK fan-in constraint.** The TT-Sparse design makes three deliberate choices regarding per-node connectivity. First, it imposes a hard top-$k$ fan-in constraint on each node before training begins, rather than relying on post-hoc sparsification of a fully connected layer. Second, it introduces

*Table 6.* Best extractable performance for each sparsification strategy, all sharing the same QMC extraction pipeline. C: rule complexity (↓). †: 3/5 seeds extractable. ‡: only extreme $\lambda{=}10$ extractable ($R^2{\approx}0$). Full Pareto grids in Appendix H.1.

| Method | Heart | | Diabetes | | Yeast | | Abalone | |
|---|---|---|---|---|---|---|---|---|
| | AUC | C | AUC | C | AUC | C | $R^2$ | C |
| TT-Sparse | **.931** | **11** | **.829** | **7** | **.826** | 110 | **.512** | 61 |
| FC + $L_0$ | .893 | 55 | .757† | 11 | .811 | 123 | −.00‡ | 1 |
| FC + $L_1$ + Prune | .902 | 155 | .676 | 17 | .798 | **96** | .152 | **3** |
| Mag. TopK | .904 | 321 | .716 | 20 | .812 | 133 | .152 | 4 |

TT-SPARSE dominates the performance-complexity trade-off on all four datasets. The $L_0$ and $L_1$ approaches, which lack a structural fan-in guarantee, can leave nodes with large or uneven fan-in that inflates extracted rule complexity even after pruning. More fundamentally, networks trained without a fan-in constraint distribute information across all available connections, learning high-order node functions. Pruning these connections post-hoc destroys the learned representations, resulting in large performance drops. In contrast, TT-SPARSE forces the network to learn compact per-node functions from the start by constraining each node to $k$ inputs throughout training. The comparison with Magnitude TopK validates the second design choice: it is not sufficient to simply select the $k$ largest LTT weights after training. Learning connectivity through a separate mapping module $W_{\text{map}}$ yields better performance-complexity tradeoffs because the mapping can specialize in selecting informative inputs independently of the logic weights that determine the Boolean function.

**Soft TopK vs. slot-based Softmax.** Given that a learned mapping is beneficial, we further validate the choice of selection mechanism. We compare against the slot-based Softmax connection selection from DWN (Bacellar et al., 2024), where instead of selecting $k$ features from a single importance vector, the $k$ inputs to each LTT node are treated as distinct, ordered "slots". The connection matrix becomes $W_{\text{map}} \in \mathbb{R}^{n \times kM}$ instead of $W_{\text{map}} \in \mathbb{R}^{n \times M}$, and each slot selects its input independently via argmax (relaxed with softmax during backpropagation). This design has two disadvantages: (1) *parameter inefficiency*, where mapping complexity scales as $\mathcal{O}(M \cdot n \cdot k)$ rather than $\mathcal{O}(M \cdot n)$; and (2) *input redundancy*, where independent slots may converge on the same feature index, effectively collapsing the dimension of the truth table. Our Soft TopK operator prevents both issues by selecting the $k$ most relevant *distinct* indices from a single importance vector per node. Experimental results (Figure 5) across all 28 datasets confirm that Soft TopK consistently yields superior predictive performance.

## 5. Conclusion

In this work, we introduced TT-SPARSE, a new neural building block designed to achieve highly performant rule sets while minimizing complexity. By reformulating truth tables as differentiable modules with sparse connectivity learned through the new soft TopK operator, we enable the end-to-end learning of Boolean logic via standard gradient descent. Furthermore, as a fully differentiable layer, TT-SPARSE offers significant architectural flexibility; it can be seamlessly integrated into standard neural pipelines to learn interpretable rules across diverse problem settings, including binary, multiclass, and regression tasks.

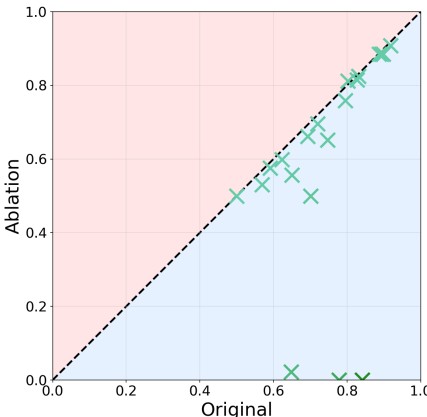

*Figure 5.* Soft TOPK vs. slot-based Softmax (Bacellar et al., 2024): AUC/$R^2$ on all 28 datasets. Points below the diagonal indicate superior performance by TT-SPARSE.

### 5.1. Limitations

Continuous features must be encoded before training, fixing the representation of the literals. Second, the utilization of the soft TOPK operator makes training reliant on the initialization of the temperature parameter $\tau$. While these parameters must be well-initialized, our ablations in Appendix H show that TT-SPARSE is robust to this constraint, maintaining high performance after as few as 5 encoding bits. Third, each LTT node is restricted to linearly separable Boolean functions of its $k$ inputs. Though it's a theoretical limitation, we validate empirically in Appendix I that this is not a practical limitation on tabular data.

### 5.2. Future Work

Future work includes extending TT-SPARSE to other data domains such as time-series via recurrent latent dimensions and supporting literals formed by learnable linear combinations of inputs (e.g. $w_1 x_1 + w_2 x_2 \leq z$) to reduce the Boolean complexity required for non-linear boundaries.

### Impact Statement

This work advances the field of interpretable machine learning by introducing a scalable framework for achieving global and exact interpretability without sacrificing predictive performance. This directly addresses the transparency requirements of high-stakes sectors such as healthcare, finance, and criminal justice, where auditability and accountability are essential. By enabling domain experts to inspect, verify, and reason about model behavior, TT-SPARSE facilitates algorithmic oversight, strengthens institutional trust, and supports informed, responsible use of machine-learning systems under emerging regulatory frameworks.

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

# A. Datasets

We benchmark with 28 publicly available datasets (14 binary, 7 multiclass, 7 regression) used in prior works (Xu et al., 2025; Grinsztajn et al., 2022; Erickson et al., 2025; Wei et al., 2019; Hegselmann et al., 2023).

*Table 7.* Dataset information: the tabular task, the dataset name, the number of rows, the number of continuous columns, categorical columns, the number of instances per class for binary and multiclass datasets.

| Task | Dataset | # Rows | # Cont. | # Cat. | Classes |
|---|---|---|---|---|---|
| Binary | bank | 45,210 | 8 | 8 | 39921/5289 |
| | blood | 748 | 4 | 0 | 570/178 |
| | calhousing | 20,640 | 8 | 0 | 10323/10317 |
| | compas | 4,966 | 3 | 8 | 2483/2483 |
| | covertype | 423,680 | 10 | 44 | 211840/211840 |
| | credit_card_default | 13,272 | 20 | 1 | 6636/6636 |
| | creditg | 1,000 | 7 | 13 | 300/700 |
| | diabetes | 768 | 8 | 0 | 500/268 |
| | electricity | 38,474 | 7 | 1 | 19237/19237 |
| | eye_movements | 7,608 | 20 | 3 | 3804/3804 |
| | heart | 918 | 5 | 6 | 410/508 |
| | income | 48,842 | 6 | 8 | 37155/11687 |
| | jungle | 44,819 | 6 | 0 | 21757/23062 |
| | road_safety | 111,762 | 29 | 3 | 55881/55881 |
| Multiclass | car | 1,728 | 0 | 6 | 1210/384/69/65 |
| | ecoli | 327 | 5 | 2 | 143/77/35/20/52 |
| | iris | 150 | 4 | 0 | 50/50/50 |
| | penguins | 333 | 4 | 3 | 146/68/119 |
| | satimage | 6,435 | 36 | 0 | 1533/703/1358/626/707/1508 |
| | wine | 178 | 13 | 0 | 59/71/48 |
| | yeast | 1,479 | 6 | 2 | 244/429/463/44/35/51/163/30/20 |
| Regression | abalone | 4,177 | 7 | 1 | - |
| | bike | 17,379 | 4 | 8 | - |
| | boston | 506 | 11 | 2 | - |
| | california | 20,640 | 8 | 0 | - |
| | crime | 1,994 | 122 | 5 | - |
| | parkinsons | 5,875 | 19 | 1 | - |
| | wine_reg | 6,497 | 11 | 0 | - |

# B. Implementation Details

## B.1. Hardware

For all experiments, we use 4 Nvidia GeForce RTX 3090 GPUs and $2\times$ Intel Xeon Silver 4310 CPUs (24 cores / 48 threads) clocked at 2.10 GHz, 128 GB RAM. Neural networks under PyTorch (TT-Sparse, NeuRules, RL-Net, RRL, DWN, TabM) were run on GPU, while tree-based model GOSDT and rule induction based models (Classy, GLRM, RuleFit) were run on CPU.

## B.2. Hyperparameters and Training Protocol

For each dataset, we reserve 20% of the data as a hold-out test set using a fixed seed. The remaining 80% is used for hyperparameter tuning. We perform a grid search by further splitting this development set into 80% training and 20% validation. Once the optimal hyperparameters are identified based on the validation metric (AUC for classification, $R^2$ for regression), we retrain the model on the full development set. This process is repeated across 5 different random seeds and we report the model's performance on the hold-out test set.

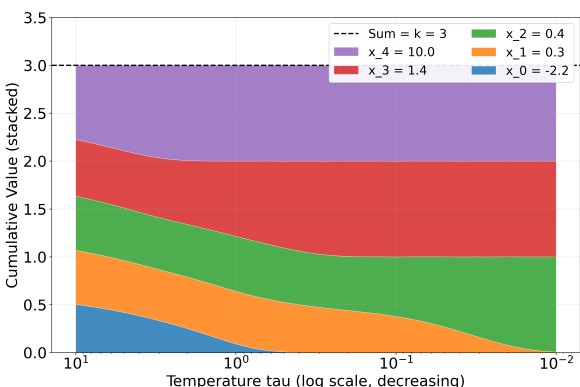

*Figure 6.* The output distribution of the soft TOPK operator with $k = 3$ applied on score vector $[-2.2, 0.3, 0.4, 1.4, 10]$ across different temperatures $\tau$. $y$-axis represents the selection probability $\pi_i$ for each index of the score vector.

All neural models (TT-Sparse, NeuRules, RL-Net, RRL, DWN) use Adam with learning rate 0.01–0.02, batch size 2048, and early stopping on validation loss. For TT-SPARSE, each continuous feature is binarized via *thermometer encoding*: we place $b$ thresholds at evenly-spaced quantiles of the training distribution and produce $b$ binary literals $\mathbf{1}(x > t_i)$ per feature. Categorical features are one-hot encoded. We tune the key architectural hyperparameters for all models by performing GridSearch:

- **TT-SPARSE:** Number of thermometer bits $b \in \{4, 5, 6\}$, Number of LTT nodes $\in \{1, 20, 30, 40, 50\}$, Soft TOPK temperature $\tau \in \{0.001, 0.01, 0.05\}$.

- **RuleFit:** max_rules $\in \{5, 20, 40, 60, 80\}$ and regularization strength $Cs \in \{1, 5, 10, 50\}$.

- **GLRM:** Regularization penalty $\lambda_0 \in \{0.05, 0.01, 0.005\}$. We fix $\lambda_1 = 0.2 \cdot \lambda_0$ as per (Wei et al., 2019).

- **Classy (MDL-Rule-List):** min_support $\in \{5, 10, 15, 20, 30\}$, maximum depth $\in \{3, 5, 7\}$, and number of cutpoints $\in \{5, 7\}$.

- **GOSDT:** Regularization parameter $\in \{0.005, 0.01, 0.05\}$ and depth budget $\in \{3, 5, 7\}$.

- **NeuRules:** n_rules $\in \{5, 10, 20, 40, 60, 80\}$, predicate temperature $\in \{0.1, 0.2\}$, and selector temperature $\in \{0.5, 1.0\}$.

- **RL-Net:** n_rules $\in \{5, 10, 20, 40, 60, 80\}$, conjunction penalty $\lambda_{and} \in \{0.01, 0.005, 0.001\}$, and $L_2$ regularization $\in \{0, 0.001, 0.01\}$.

- **DWN (Linear):** Hidden layer sizes $\in \{20, 30, 40, 50\}$ and Look-Up Table (LUT) size $\in \{4, 5, 6\}$. To ensure a fair comparison with TT-SPARSE, we implemented the classifier layer immediately following the LUT layer. This architecture introduces a linear operation by assigning a weight to each LUT and adding a bias term before the activation function. The logic rules are derived similarly as TT-SPARSE, utilizing Quine-McCluskey conversion to transform LUT minterms into DNF equations.

- **TabM**: We use the AdamW optimizer and a batch size of 512. Number of layers $\in \{1, 2, 3\}$, hidden size $\in \{64, 128, 256, 512\}$, dropout rate $\in \{0, 0.1, 0.2\}$, learning rate $\in \{0.01, 0.005, 0.001\}$, bin count for embeddings $\in \{8, 16, 32, 64\}$, and embedding dimension $\in \{8, 16, 32, 64\}$.

## C. Soft TOPK

$S_k(\vec{s}) = [\pi_1, \dots, \pi_n]$ where $\pi_i = \sigma(\frac{s_i}{\tau} + c)$, constrained by $\sum_{j=1}^{n} \sigma(\frac{s_j}{\tau} + c) = k$. $\sigma(x) = \frac{1}{1+e^{-x}}$ is the sigmoid function and $\sigma'(x) = \sigma(x)(1 - \sigma(x))$. Let $d_i = \sigma'(\frac{s_i}{\tau} + c) = \frac{d}{d(\frac{s_i}{\tau}+c)}\sigma(\frac{s_i}{\tau} + c) = \pi_i(1 - \pi_i)$ and $\vec{d} = [d_1, \dots, d_n]^\top$.

### C.1. Solving the optimization objective via Lagrange multipliers

Recall the optimization objective

$$\vec{\pi}^* = \underset{\vec{\pi} \in [0,1]^n}{\operatorname{argmax}} \left[ \sum_{i=1}^{n} \pi_i s_i + \tau \sum_{i=1}^{n} H(\pi_i) \right] \text{ s.t. } \sum_{i=1}^{n} \pi_i = k \tag{10}$$

where $H(\pi_i)$ is the binary entropy $H(\pi_i) = -\pi_i \log \pi_i - (1 - \pi_i) \log(1 - \pi_i)$. We then derive the optimum by introducing a Lagrange multiplier $\lambda$ for the $\sum_{i=1}^{n} \pi_i = k$ constraint such that the problem becomes

$$\mathcal{L}(\vec{\pi}, \lambda) = \sum_{i=1}^{n} \pi_i s_i + \tau \sum_{i=1}^{n} \left( -\pi_i \log \pi_i - (1-\pi_i) \log(1-\pi_i) \right) - \lambda \left( \sum_{i=1}^{n} \pi_i - k \right) \tag{11}$$

$$\frac{\partial \mathcal{L}}{\partial \pi_i} = s_i + \tau \left( \log \frac{1-\pi_i}{\pi_i} \right) - \lambda = 0 \tag{12}$$

Setting the derivative 0, we obtain the solution $\pi_i = \frac{1}{1 + e^{-\frac{(s_i - \lambda)}{\tau}}} = \sigma(\frac{s_i - \lambda}{\tau})$, where $\sigma(\cdot)$ is the sigmoid function. Thus, setting $c = -\frac{\lambda}{\tau}$, we define $S_k(\vec{s}) := \vec{\pi}$ where $\pi_i := \sigma(\frac{s_i}{\tau} + c)$ for $i \in \{1, \ldots, n\}$ and $c \in \mathbb{R}$ is the unique root of $f(c) = \sum_{j=1}^{n} \sigma(\frac{s_j}{\tau} + c) - k = 0$. Since $f(c)$ is strictly increasing, $c$ is uniquely determined and solved by the bisection method.

### C.2. Obtaining $\nabla_{\vec{s}} S_k(\vec{s})$

We want the Jacobian $J \in R^{n \times n}$ where $J_{ij} = \frac{\partial \pi_i}{\partial s_j}$.

First, we apply the chain rule to $\pi_i = \sigma(\frac{s_i}{\tau} + c)$:

$$\frac{\partial \pi_i}{\partial s_j} = \sigma'\left(\frac{s_i}{\tau} + c\right) \cdot \frac{\partial}{\partial s_j}\left(\frac{s_i}{\tau} + c\right)$$

Since $\frac{\partial s_i}{\partial s_j} = \delta_{ij}$:

$$\frac{\partial \pi_i}{\partial s_j} = d_i \left( \frac{1}{\tau} \delta_{ij} + \frac{\partial c}{\partial s_j} \right) \tag{13}$$

where $\delta_{ij} = \mathbf{1}_{\{i=j\}}$ is the Kronecker delta. Next, we differentiate the constraint function w.r.t. $s_j$ to obtain:

$$\frac{\partial}{\partial s_j} \sum_{m=1}^{n} \sigma\left(\frac{s_m}{\tau} + c\right) = 0. \tag{14}$$

$$\sum_{m=1}^{n} d_m \left( \frac{1}{\tau} \delta_{mj} + \frac{\partial c}{\partial s_j} \right) = 0. \tag{15}$$

$$\frac{1}{\tau} d_j + \left( \sum_{m=1}^{n} d_m \right) \frac{\partial c}{\partial s_j} = 0. \tag{16}$$

$$\frac{\partial c}{\partial s_j} = -\frac{1}{\tau} \frac{d_j}{\|\vec{d}\|_1} \tag{17}$$

Substituting this back to Equation (13):

$$\frac{\partial \pi_i}{\partial s_j} = \frac{1}{\tau} d_i \left( \delta_{ij} - \frac{d_j}{\|\vec{d}\|_1} \right) \tag{18}$$

$$\nabla_{\vec{s}} S_k(\vec{s}) = \frac{1}{\tau} \left[ \operatorname{diag}(\vec{d}) - \frac{\vec{d}\,\vec{d}^\top}{\|\vec{d}\|_1} \right] \tag{19}$$

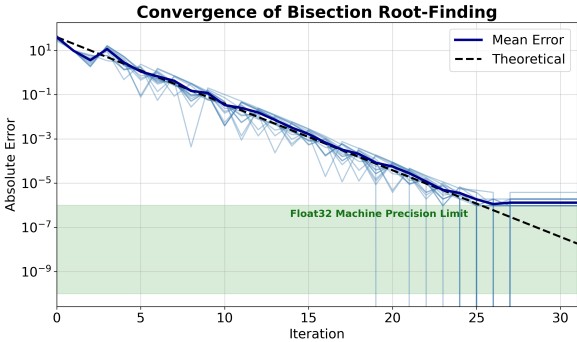

*Figure 7.* Absolute error $|\sum \pi_i - k|$ (log scale) vs. iterations for a batch of 16 score vectors $\vec{s} \in \mathbb{R}^{100}$ with target $k = 10$. Each light blue line represents the error plot of a score vector in the batch.

## C.3. Obtaining $\nabla_k S_k(\vec{s})$

$$\frac{\partial \pi_i}{\partial k} = d_i \cdot \frac{\partial}{\partial k} \left( \frac{s_i}{\tau} + c \right) \tag{20}$$

$$\frac{\partial \pi_i}{\partial k} = d_i \cdot \frac{\partial c}{\partial k} \tag{21}$$

Then, similarly, we differentiate both sides of the constraint function w.r.t. $k$ to obtain:

$$\frac{\partial}{\partial k} \sum_{m=1}^{n} \sigma \left( \frac{s_m}{\tau} + c \right) = 1 \tag{22}$$

$$\sum_{m=1}^{n} d_m \cdot \frac{\partial c}{\partial k} = 1 \tag{23}$$

$$\frac{\partial c}{\partial k} = \frac{1}{\|\vec{d}\|_1} \tag{24}$$

Substituting this back to Equation (17):

$$\frac{\partial \pi_i}{\partial k} = \frac{d_i}{\|\vec{d}\|_1} \tag{25}$$

$$\nabla_k S_k(\vec{s}) = \frac{\vec{d}}{\|\vec{d}\|_1} \tag{26}$$

## C.4. Convergence of bisection method

To compute the constant $c$, we implement vectorized bisection search that solves for $c$ across an entire batch simultaneously on GPU. The bisection method halves the search space in every iteration, so the absolute error $|\sum S_k(\vec{s})_i - k|$ decays with $\epsilon_t \propto 2^{-t}$.

Figure 7 illustrates the convergence of this method. The soft TOPK is applied to a batch of 16 score vectors $\vec{s} \in \mathbb{R}^{100}$ with target $k = 10$. The theoretical linear convergence (linear slope on the semi-log plot) is observed empirically. The error magnitude drops below $10^{-3}$ within 15 iterations and reaches the limit of `Float32` machine precision in around 30 iterations. Our experiments also show negligible overhead with around 1 ms for a batch of size 16, giving a throughput of more than 10000 samples per second.

## D. TT-SPARSE Parameter Gradients

Let $\vec{x} \in \mathbb{R}^n$ denote the input to the TT-SPARSE layer with $M$ LTT nodes. The layer is parameterized by $W_{\text{map}} \in \mathbb{R}^{n \times M}$, $W_{\text{LTT}} \in \mathbb{R}^{n \times M}$, and $b \in \mathbb{R}^M$. For each node $j \in \{1, \dots, M\}$, the output is

$$z_j = \sum_{i=1}^{n} m_i^{(j)} \cdot W_{\text{LTT}}[i,j] \cdot x_i + b_j.$$

Leveraging the soft TOPK operator $S_k$ in the backpropagation, $m^{(j)} = S_k(W_{\text{map}}[\cdot, j])$ is the operator applied on the $j$-th column of $W_{\text{map}}$ (i.e., $\vec{s} = W_{\text{map}}[\cdot, j]$) which corresponds to the $j$-th node. Let $\mathcal{L}$ be the upstream loss.

### D.1. Gradient w.r.t. $W_{\text{map}}$

Applying the chain rule and denoting the Jacobian $J_j = \frac{\partial S_k(W_{\text{map}}[\cdot,j])}{\partial W_{\text{map}}[\cdot,j]}$, we have

$$\frac{\partial \mathcal{L}}{\partial W_{\text{map}}[\cdot,j]} = J_j^\top \left( \frac{\partial \mathcal{L}}{\partial z_j} \cdot \frac{\partial z_j}{\partial S_k(W_{\text{map}}[\cdot,j])} \right)$$

As $m^{(j)} = S_k(W_{\text{map}}[\cdot, j])$, we have

$$\frac{\partial z_j}{\partial S_k(W_{\text{map}}[\cdot,j])} = \vec{x} \odot W_{\text{LTT}}[\cdot, j]. \tag{27}$$

The Jacobian $J_j$ is that of the soft TOPK operator applied on the $j$-th column of $W_{\text{map}}$. Using the Jacobian formula obtained in Equation (5), where $\vec{d} = m^{(j)} \odot (1 - m^{(j)})$ is the elementwise sigmoid derivative evaluated at the soft TOPK output:

$$\frac{\partial S_k(W_{\text{map}}[\cdot,j])}{\partial W_{\text{map}}[\cdot,j]} = \frac{1}{\tau} \left[ \text{diag}(\vec{d}) - \frac{\vec{d}\vec{d}^\top}{\|\vec{d}\|_1} \right] \tag{28}$$

Let $g_j = \frac{\partial \mathcal{L}}{\partial z_j} \cdot (\vec{x} \odot W_{\text{LTT}}[\cdot, j])$ be the local gradient from the first and second terms of the derivative. Since $J_j$ is symmetric, we obtain the vector-Jacobian product:

$$\frac{\partial \mathcal{L}}{\partial W_{\text{map}}[\cdot,j]} = \frac{1}{\tau} \left[ \text{diag}(\vec{d}) g_j - \frac{\vec{d}\vec{d}^\top}{\|\vec{d}\|_1} g_j \right] = \frac{1}{\tau} \left[ \vec{d} \odot g_j - \frac{\vec{d}(\vec{d} \cdot g_j)}{\|\vec{d}\|_1} \right] \tag{29}$$

### D.2. Gradient w.r.t. $W_{\text{LTT}}$ and $b$

$$\frac{\partial \mathcal{L}}{\partial W_{\text{LTT}}[i,j]} = \frac{\partial \mathcal{L}}{\partial z_j} \cdot \frac{\partial z_j}{\partial W_{\text{LTT}}[i,j]} = \frac{\partial \mathcal{L}}{\partial z_j} \cdot m_i^{(j)} \cdot \vec{x}_i \tag{30}$$

$$\frac{\partial \mathcal{L}}{\partial b_j} = \frac{\partial \mathcal{L}}{\partial z_j} \cdot \frac{\partial z_j}{\partial b_j} = \frac{\partial \mathcal{L}}{\partial z_j} \tag{31}$$

# E. Performance-Complexity Pareto Grids

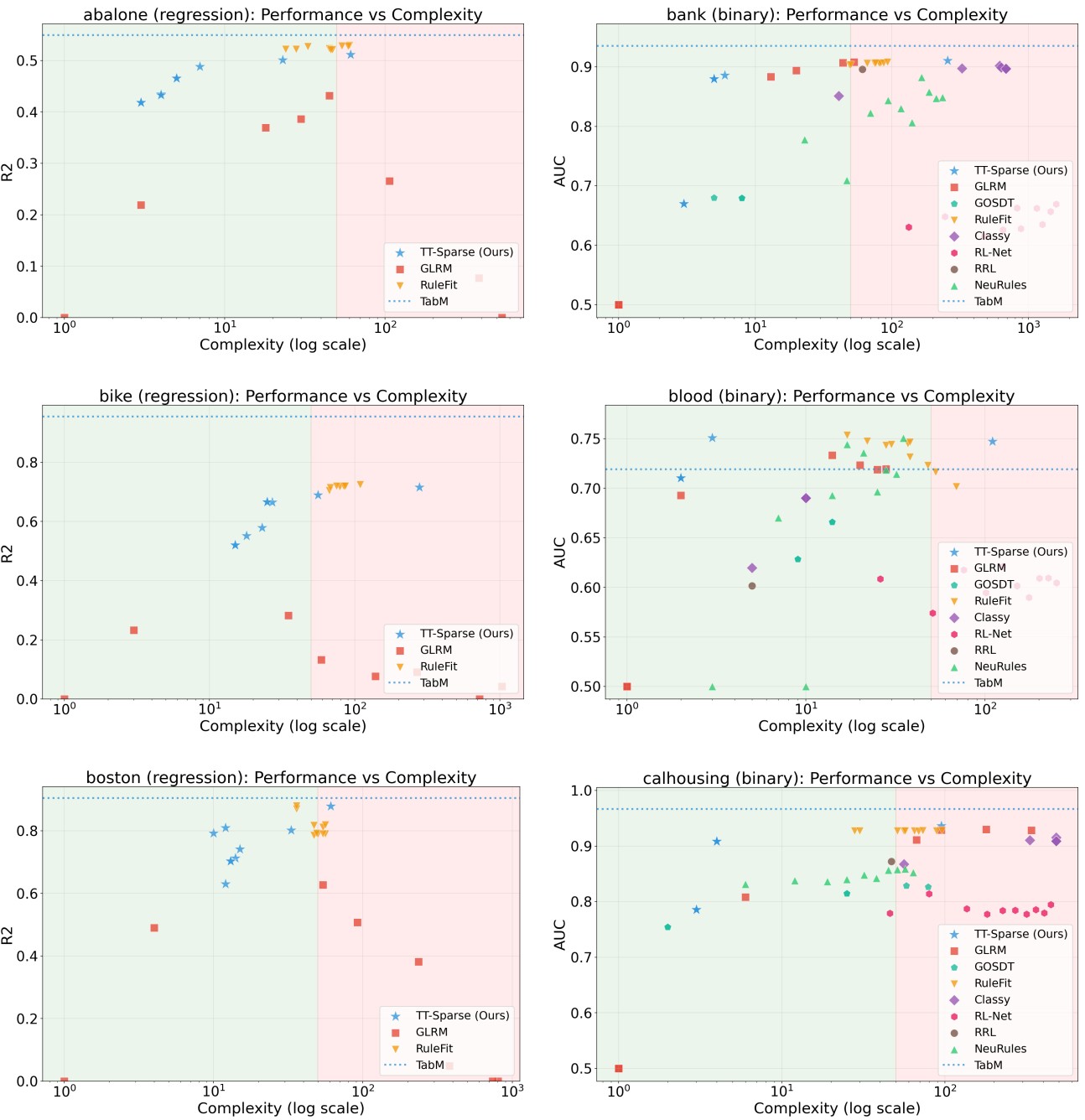

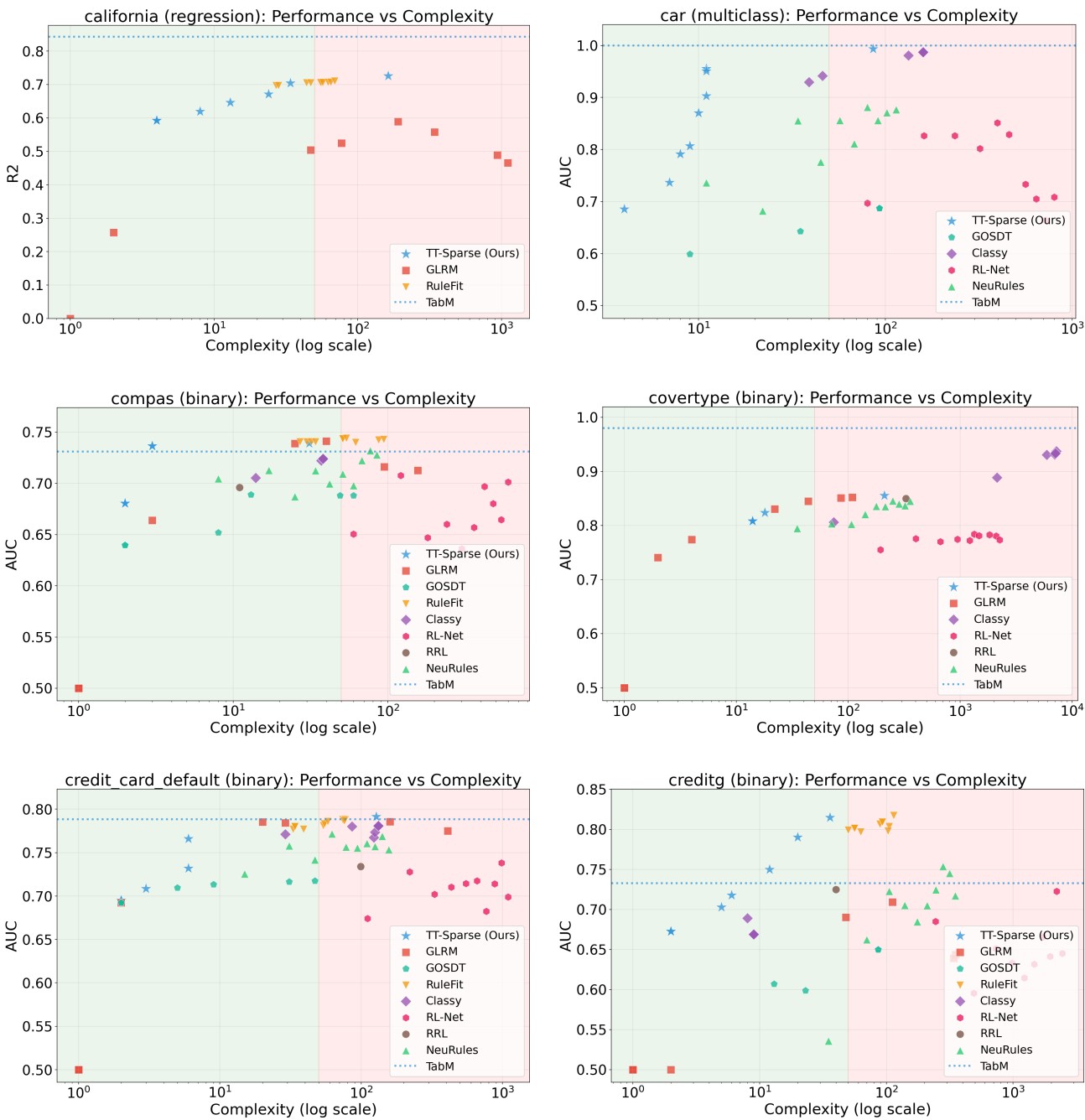

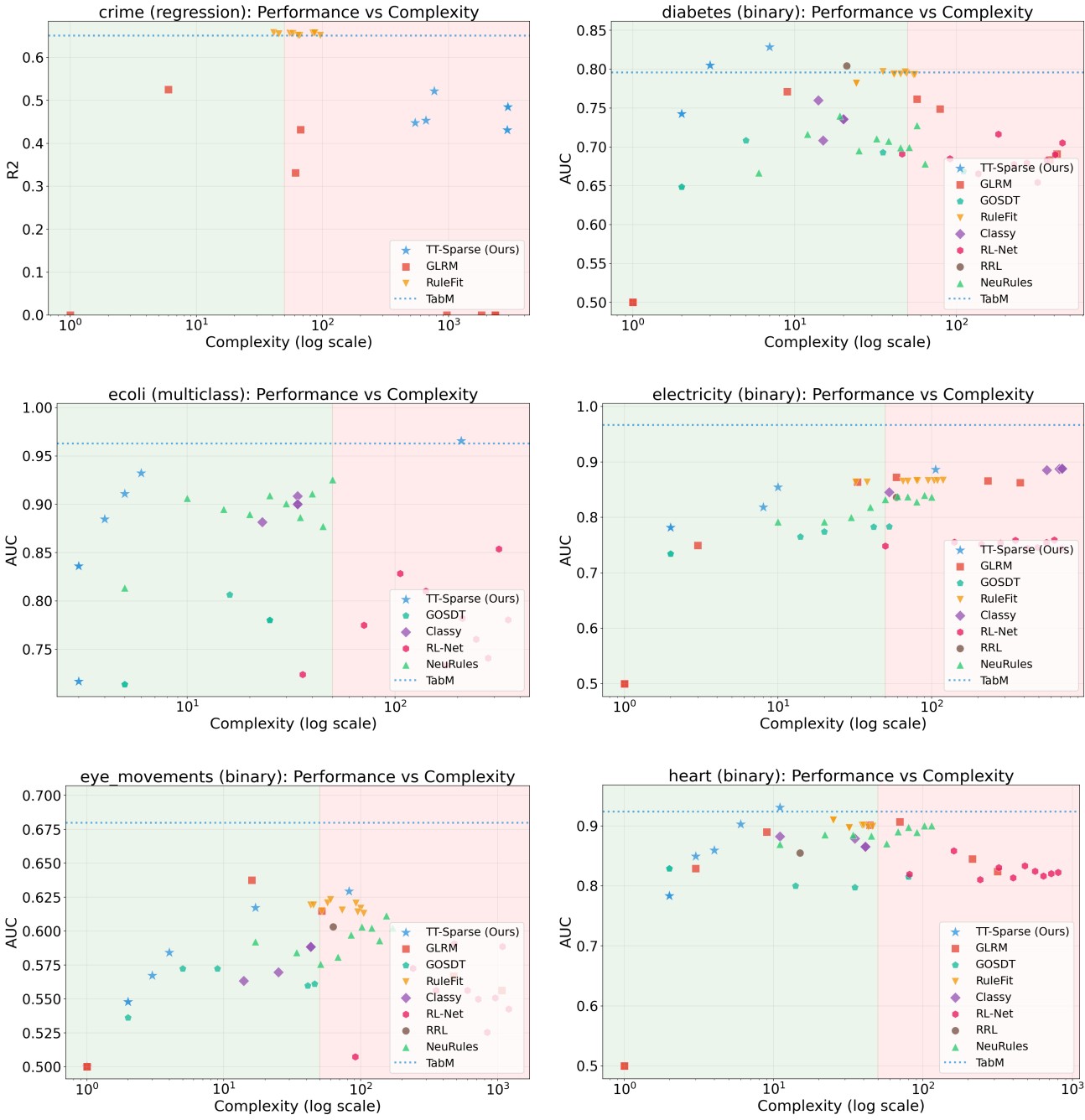

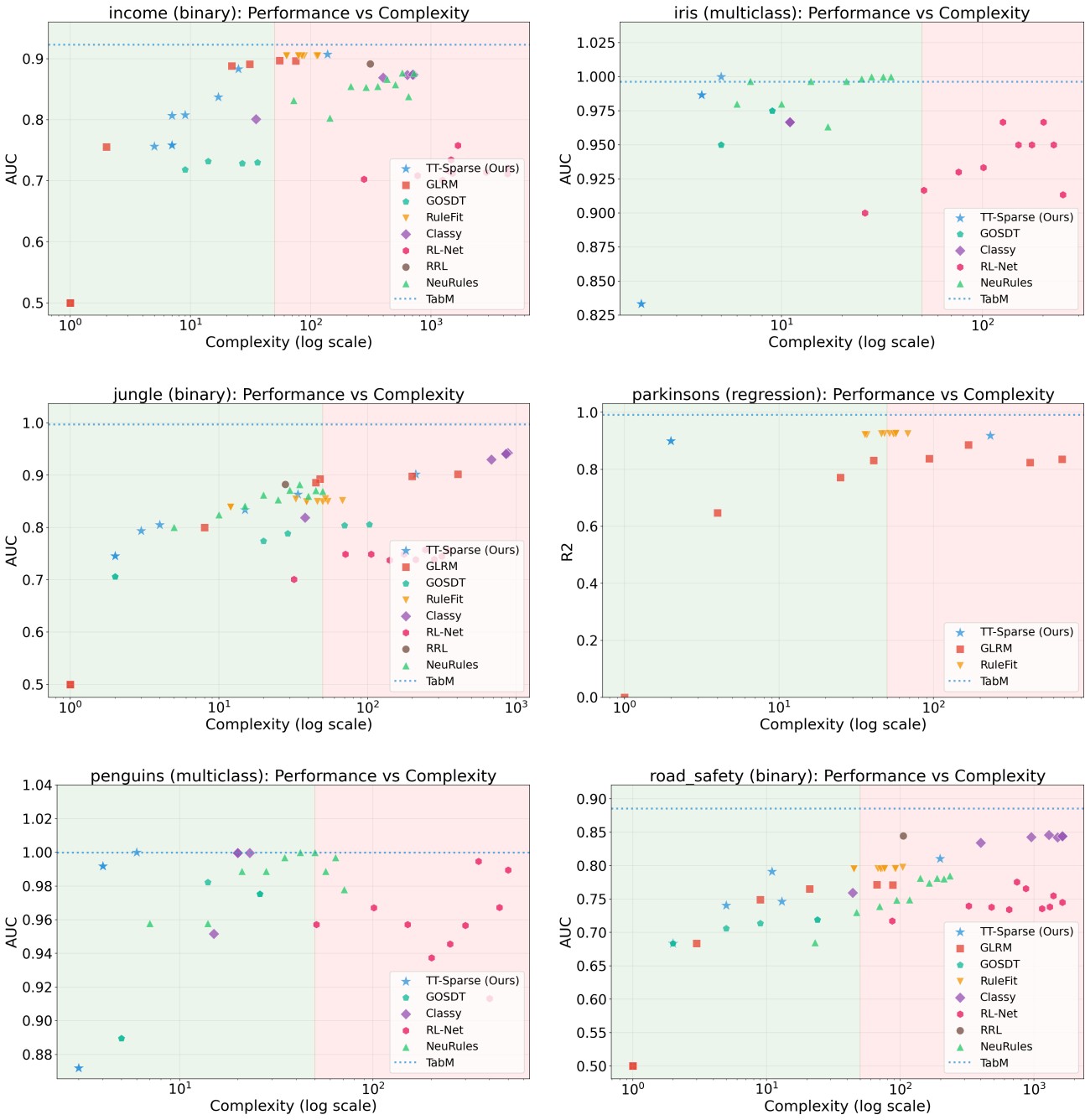

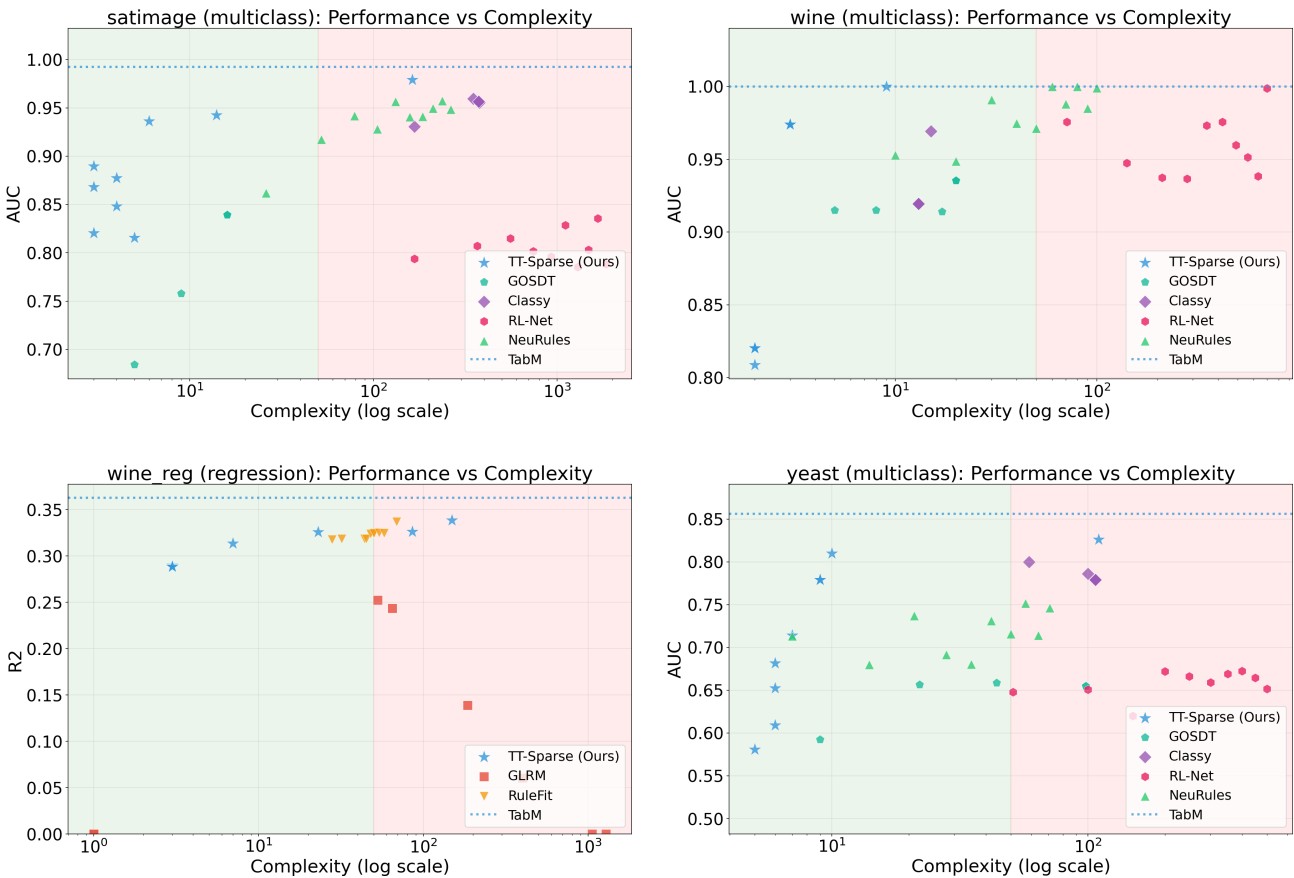

## F. Example rules

**GLRM** example rule for eye_movements dataset with complexity 15:
**0.84** – titleNo $\leq$ 2;   **0.34** – wordNo $\leq$ 2;   **-0.33** – nextWordRegress $=$ 1;   **0.31** – regressDur $\leq$ 139;   **0.13** – nextWordRegress $=$ 0;   **-0.13** – wordNo $\leq$ 1;   **0.15** – regressDur $\leq$ 0;   **-0.18** – prevFixPos $\leq$ 149.5;   **-0.16** – landingPos $\leq$ 71.6;   **-0.10** – firstSaccLen $\leq$ 250.1;   **-0.04** – firstSaccLen $\leq$ 204.0;   **-0.53** – (lastSaccLen $\leq$ 503.2 $\wedge$ landingPos $\leq$ 117.3 $\wedge$ 1 $<$ wordNo $\leq$ 8);   (Bias: -0.23)

**GOSDT** example rule in the form of a tree for blood dataset with complexity 14:
**Class 0** – Feature$_0$ $>$ $-0.62$;   **Class 0** – Feature$_0$ $\leq$ $-0.62 \wedge$ Feature$_3$ $>$ 0.34;   **Class 1** – Feature$_0$ $\leq$ $-0.62 \wedge$ Feature$_3$ $\leq$ 0.34 $\wedge$ Feature$_6$ $>$ 2.14;   **Class 0** – Feature$_0$ $\leq$ $-0.62 \wedge$ Feature$_3$ $\leq$ 0.34 $\wedge$ Feature$_6$ $\leq$ 2.14 $\wedge$ Feature$_8$ $>$ 0.34;   **Class 1** – Feature$_0$ $\leq$ $-0.62 \wedge$ Feature$_3$ $\leq$ 0.34 $\wedge$ Feature$_6$ $\leq$ 2.14 $\wedge$ Feature$_8$ $\leq$ 0.34;

**Classy (MDL Rule List)** example decision list for blood dataset with complexity 8:
**Class 1** ($P = 0.57$) – (monetary $\geq$ 1500 $\wedge$ time $<$ 59 $\wedge$ recency $<$ 7);   **Class 0** ($P = 0.71$) – recency $<$ 7;   **Default Class 0** ($P = 0.89$) – else.

**NeuRules** example rule for diabetes dataset with complexity 17:
**3.19** – (0.82 $<$ Feature_0 $<$ 1.69) $\rightarrow$ Class 1(71.5%);   **8.41** – (0.46 $<$ Feature_4 $\wedge$ $-0.40$ $<$ Feature_7 $<$ 3.50) $\rightarrow$ Class 1(74.2%);   **4.56** – (0.62 $<$ Feature_4 $\wedge$ $-0.15$ $<$ Feature_7 $<$ 3.31) $\rightarrow$ Class 1(62.8%);   **5.97** – (Feature_0 $<$ 2.49 $\wedge$ Feature_1 $<$ 0.58 $\wedge$ Feature_6 $<$ 4.55) $\rightarrow$ Class 0(80.9%);   **1.00** – ($-3.49$ $<$ Feature_1 $<$ 1.67 $\wedge$ 0.18 $<$ Feature_4 $<$ 4.98 $\wedge$ $-1.14$ $<$ Feature_6 $<$ 5.58 $\wedge$ $-0.35$ $<$ Feature_7 $<$ 3.50) $\rightarrow$ Class 1(64.4);

**RL-Net** example rule for calhousing dataset with complexity 87:
**Class 1** – (Feature$_0$ $\wedge$ $\neg$Feature$_4$ $\wedge$ $\neg$Feature$_6$ $\wedge$ $\neg$Feature$_7$);   **Class 1** – ($\neg$Feature$_0$ $\wedge$ Feature$_1$ $\wedge$ $\neg$Feature$_2$ $\wedge$ Feature$_3$ $\wedge$ $\neg$Feature$_4$ $\wedge$ Feature$_5$ $\wedge$ $\neg$Feature$_6$ $\wedge$ $\neg$Feature$_7$);   **Class 1** – ($\neg$Feature$_0$ $\wedge$ Feature$_1$ $\wedge$ Feature$_2$ $\wedge$ Feature$_3$ $\wedge$ $\neg$Feature$_4$ $\wedge$ Feature$_5$ $\wedge$ $\neg$Feature$_6$ $\wedge$ $\neg$Feature$_7$);   **Class 1** – (Feature$_0$ $\wedge$ $\neg$Feature$_1$ $\wedge$ Feature$_2$ $\wedge$ Feature$_3$ $\wedge$ Feature$_4$ $\wedge$ Feature$_5$ $\wedge$ $\neg$Feature$_6$ $\wedge$ $\neg$Feature$_7$);   **Class 0** – ($\neg$Feature$_0$ $\wedge$ Feature$_1$ $\wedge$ $\neg$Feature$_2$ $\wedge$ Feature$_3$ $\wedge$ Feature$_4$ $\wedge$ Feature$_5$ $\wedge$ Feature$_6$ $\wedge$ Feature$_7$);   **Class 0** – ($\neg$Feature$_0$ $\wedge$ Feature$_1$ $\wedge$ $\neg$Feature$_2$ $\wedge$ $\neg$Feature$_3$ $\wedge$ Feature$_4$ $\wedge$ Feature$_5$ $\wedge$ $\neg$Feature$_6$ $\wedge$ Feature$_7$);   **Class 0** – ($\neg$Feature$_0$ $\wedge$ Feature$_1$ $\wedge$ $\neg$Feature$_2$ $\wedge$ Feature$_3$ $\wedge$ $\neg$Feature$_4$ $\wedge$ Feature$_5$ $\wedge$ $\neg$Feature$_6$ $\wedge$ Feature$_7$);   **Class 0** – ($\neg$Feature$_0$ $\wedge$ Feature$_1$ $\wedge$ $\neg$Feature$_2$ $\wedge$ Feature$_3$ $\wedge$ Feature$_4$ $\wedge$ Feature$_5$ $\wedge$ $\neg$Feature$_6$ $\wedge$ $\neg$Feature$_7$);   Default: **Class 0**

**RRL** example rule with complexity 12:
**0.12** – frequency $>$ 17.85;   **-0.05** – frequency $\leq$ 3.14;   **0.04** – recency $>$ 19.73;   **0.04** – (recency $>$ 32.27 $\vee$ time $\leq$ 31.02);   **0.19** – (recency $>$ 12.79 $\wedge$ time $>$ 10.96 $\wedge$ frequency $\leq$ 1.75);   **0.09** – (time $>$ 16.72 $\wedge$ frequency $\leq$ 2.10 $\wedge$ monetary $\leq$ 1131.57);   *(Bias: -0.02)*

**RuleFit** example rule on abalone dataset with complexity 15:
**12.63** – Feature_5;   **-11.86** – Feature_6;   **-3.28** – Feature_7;   **3.12** – Feature_8;   **1.83** – Feature_4;   **1.36** – Feature_3;   **-0.74** – Feature_0;   **-0.12** – Feature_2;   **0.08** – Feature_1;   **-0.30** – (Feature_8 $\leq$ $-0.62$);   **0.09** – (Feature_8 $>$ $-0.64 \wedge$ Feature_8 $\leq$ 0.83);   **-0.43** – (Feature_8 $>$ $-0.64 \wedge$ Feature_8 $>$ 0.83);   (Bias: 10.24)

**TT-SPARSE** example rule (same as Figure 2) with complexity 15:
**-0.83** – Feature_ChestPainType $=$ 'NAP';   **0.97** – Feature_ExerciseAngina $=$ 'Y';   **1.13** – Feature_ST_Slope $=$ 'Flat';   **1.40** – ((Feature_Cholesterol $<$ 167.63 $\wedge$ Feature_Oldpeak $\geq$ 3.00) $\vee$ (Feature_ChestPainType $\neq$ 'TA' $\wedge$ Feature_Oldpeak $\geq$ 3.00) $\vee$ (Feature_ChestPainType $\neq$ 'TA' $\wedge$ Feature_ChestPainType $\neq$ 'ATA' $\wedge$ Feature_Cholesterol $<$ 167.63));   **-1.03** – ((Feature_MaxHR $\geq$ 177.25 $\wedge$ Feature_Oldpeak $<$ 2.25) $\vee$ (Feature_ChestPainType $=$ 'ATA' $\wedge$ Feature_Cholesterol $<$ 224.88));   (Bias: -1.04)

**TT-SPARSE** example multiclass rule (Iris dataset) with complexity 7: **Class 0: -3.66** – sepal-length;   **Class 0: 7.10, Class 2: -2.91** – sepal-width;   **Class 0: -9.47, Class 2: 6.42** – petal-length;   **Class 0: -10.65, Class 1: -4.34, Class 2: 5.77** – petal-width;   **Class 1: -10.55, Class 2: 8.85** – (petal-length $\geq$ 5.76 $\vee$ petal-width $\geq$ 1.87);   (Bias: **Class 0: 2.70, Class 1: 8.71, Class 2: -2.82**)

## G. Quine-McCluskey

---

**Algorithm 1** Quine-McCluskey with XOR/XNOR

---

**Require:** Set of minterms $M_1$, set of don't-cares $M_{dc}$, flag $use\_xor$
**Ensure:** Minimal set of implicants $R$ covering $M_1$
 1: **procedure** SIMPLIFY($M_1, M_{dc}$)
 2:     $T \leftarrow$ BinaryStringRep($M_1 \cup M_{dc}$)
 3:     $PI \leftarrow$ GETPRIMEIMPLICANTS($T, use\_xor$)
 4:     $EI \leftarrow$ EXTRACTESSENTIAL($PI, M_{dc}$)
 5:     $R \leftarrow$ REDUCEIMPLICANTS($EI, M_{dc}$)
 6:     **return** $R$
 7: **end procedure**

 8: **function** GETPRIMEIMPLICANTS($T, use\_xor$)
 9:     **if** $use\_xor$ **then**
10:         $T \leftarrow T \cup \{$MergeXOR($t_i, t_j$) $\mid t_i, t_j \in T\}$ {Pre-pass for simple XOR/XNOR}
11:     **end if**
12:     $marked \leftarrow \emptyset$
13:     **while** changes occur in $T$ **do**
14:         Group $T$ by tuple $(n_{\text{ones}}, n_\oplus, n_\odot)$
15:         $T_{new} \leftarrow \emptyset$
16:         **for** each group $G_k$ and adjacent group $G_{next}$ **do**
17:             **for** $t_1 \in G_k, t_2 \in G_{next}$ **do**
18:                 **if** $d_H(t_1, t_2) = 1$ **then** {Standard QM Merge}
19:                     $t_{new} \leftarrow$ ReplaceDiff($t_1, t_2,$ '-')
20:                     $T_{new} \leftarrow T_{new} \cup \{t_{new}\}$; $marked \leftarrow marked \cup \{t_1, t_2\}$
21:                 **else if** $use\_xor \wedge$ IsComplement($t_1, t_2$) **then** {Extended Merge}
22:                     $t_{new} \leftarrow$ ReplaceDiff($t_1, t_2, \oplus$ or $\odot$)
23:                     $T_{new} \leftarrow T_{new} \cup \{t_{new}\}$; $marked \leftarrow marked \cup \{t_1, t_2\}$
24:                 **end if**
25:             **end for**
26:         **end for**
27:         $PI \leftarrow PI \cup (T \setminus marked)$
28:         $T \leftarrow T_{new}$
29:     **end while**
30:     **return** $PI$
31: **end function**

32: **function** REDUCEIMPLICANTS($I, M_{dc}$)
33:     **Def** $E(t)$: Set of minterms covered by term $t$ excluding $M_{dc}$
34:     **Def** $C(t)$: Complexity cost of term $t$ (weighted sum of operators)
35:     {Step 1: Orthogonal Combination}
36:     **repeat**
37:         Find pair $a, b \in I$ and merger $m$ such that $E(m) = E(a) \cup E(b)$
38:         **if** exists $m$ **then**
39:             $I \leftarrow (I \setminus \{a, b\}) \cup \{m\}$
40:         **end if**
41:     **until** no combinations found
42:     {Step 2: Redundancy Elimination}
43:     **repeat**
44:         Find $t \in I$ such that $E(t) \subseteq \bigcup_{k \in I \setminus \{t\}} E(k)$
45:         **if** exists $t$ **then**
46:             Select $t_{worst} \in \{t \mid t$ is redundant$\}$ maximizing $C(t)$
47:             $I \leftarrow I \setminus \{t_{worst}\}$
48:         **end if**
49:     **until** no redundant terms exist
50:     **return** $I$
51: **end function**

# H. Ablation Studies

We conduct ablation studies to isolate the impact of specific components and hyperparameters within the TT-SPARSE framework. In Figure 13, we analyze the effect of the number of thermometer bits $b$ (i.e., the number of quantile-based thresholds per continuous feature) on both predictive performance and rule complexity. We observe a performance plateau beyond 7 bits, indicating that TT-SPARSE effectively captures decision boundaries without requiring finer quantization. Figures 14 and 15 examine the model's sensitivity to key structural parameters: the sparsity degree (number of input bits per LTT node), the model capacity (number of LTT nodes), and the temperature $\tau$ of the Soft TOPK operator. Finally, we validate the necessity of our hybrid design by ablating the global skip connections that link input features directly to the final classifier. We observe that removing this skip connection degrades performance.

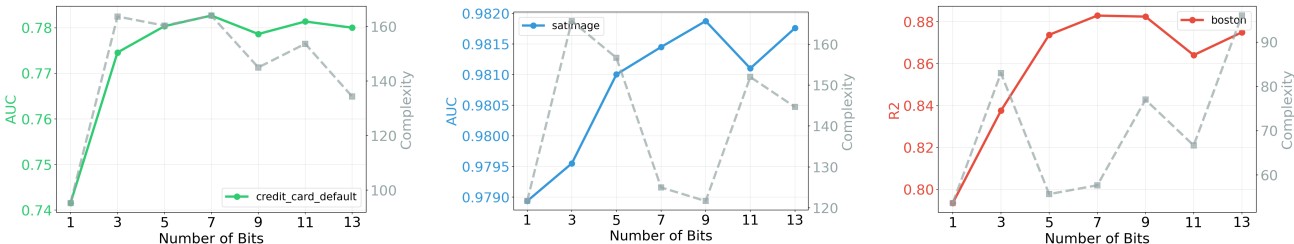

*Figure 13.* Ablation study on the number of bits used for continuous feature encoding, showing the performance metric (AUC or $R^2$) on the left side of the y-axis and rule complexity (in gray) on the right.

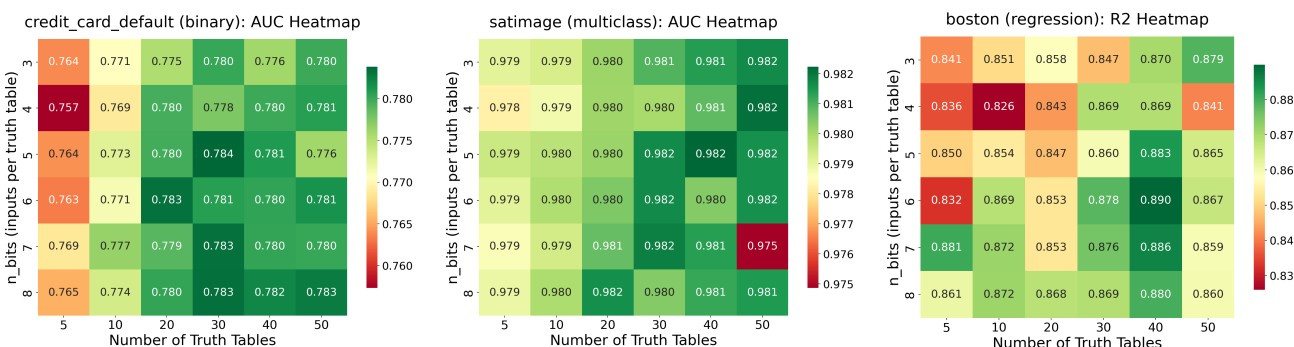

*Figure 14.* Ablation study on the number of input bits into each LTT node and the number of LTT nodes in the layer, visualized with a heatmap of the performance metric (AUC or $R^2$).

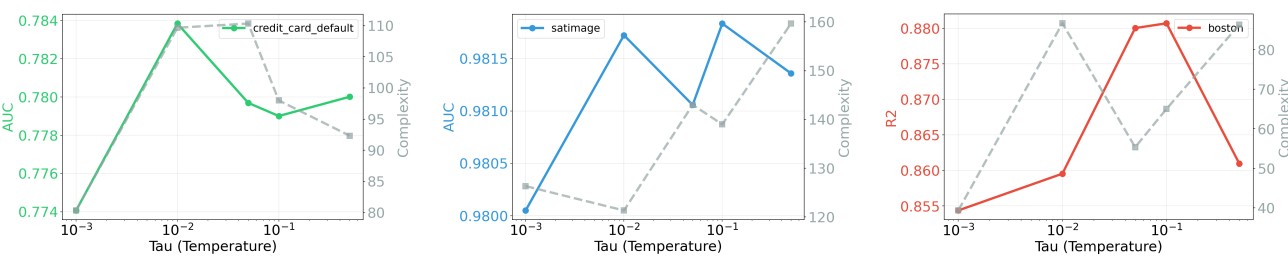

*Figure 15.* Ablation study on the temperature $\tau$ of the soft TOPK operator (lower $\tau$ gets it closer to discrete TOPK).

## H.1. TopK Necessity: Full Pareto Grids

Figure 17 shows the full performance–complexity Pareto frontiers for TT-SPARSE and the three generic sparsification baselines from Section 4.1. Each point represents one $(\lambda, \text{seed})$ configuration. TT-SPARSE consistently occupies the upper-left region (high performance, low complexity), while the baselines are pushed toward the right (high complexity) or bottom (low performance). Notably, FC + $L_0$ gates on Abalone produces no extractable points at moderate $\lambda$, and collapses to trivial predictions ($R^2 \approx 0$) at extreme $\lambda$.

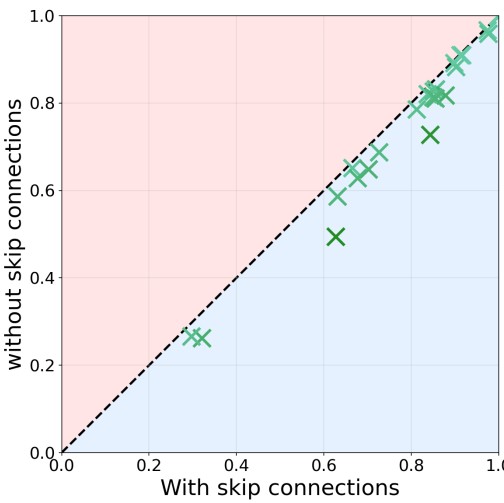

*Figure 16.* Ablation study on the integration of skip connection from the input features directly to the final classifier, concatenated with the outputs of the TT-SPARSE block. Each points show the evaluation metric (AUC/R2) achieved by the model with and without skip connections.

# I. Expressivity Validation: Deep LTT

### I.1. Architecture

Each standard LTT node computes a linear threshold function $y_j(v) = \mathbf{1}(v^\top w^{(j)} + b_j > 0)$, which can only represent linearly separable Boolean functions. To test whether this restriction limits performance, we introduce a *deep LTT* variant that achieves universal Boolean expressivity per node. The architecture replaces the single linear combination with a two-path computation (Figure 18):

1. **Hidden path (nonlinear):** The $k$ selected inputs pass through a hidden layer with $H$ units ($H = 2^k$ by default) and ReLU activation, followed by a linear output projection. This path can approximate any Boolean function given sufficient hidden units.

2. **Skip path (linear):** A direct linear combination of the $k$ inputs (identical to the standard LTT computation), providing a residual connection.

The node output combines both paths: $z_j(x) = \underbrace{h(x)^\top w_{\text{out}}^{(j)}}_{\text{hidden path}} + \underbrace{x_{\mathcal{I}_j}^\top w_{\text{skip}}^{(j)}}_{\text{skip path}} + b_j$, binarized as before via $y_j = \mathbf{1}(z_j > 0)$. The

same Soft TOPK connection selection operates on both paths simultaneously through a shared mask.

### I.2. Results

**Synthetic expressivity.** We train single-node models on all $2^{2^3} = 256$ possible 3-bit Boolean functions. Deep LTT achieves 100% exact fit rate while the standard LTT fits only 40.6% (the linearly separable subset), confirming that the theoretical gap is real.

**Real-world evaluation.** We compare both variants across 27 datasets (14 binary, 6 multiclass, 7 regression) with full hyperparameter search and 5 seeds. Table 8 summarizes results by task type. The standard LTT wins on 24/27 datasets by predictive metric; deep LTT wins on only 3. Mean AUC/$R^2$ deltas are consistently negative (deep minus standard), indicating the additional expressivity does not translate to gains on tabular data.

The linear threshold inductive bias acts as beneficial regularization: it constrains each node toward simple decision boundaries, which aligns with the structure of real tabular data. The standard LTT is therefore the appropriate default for interpretable rule learning.

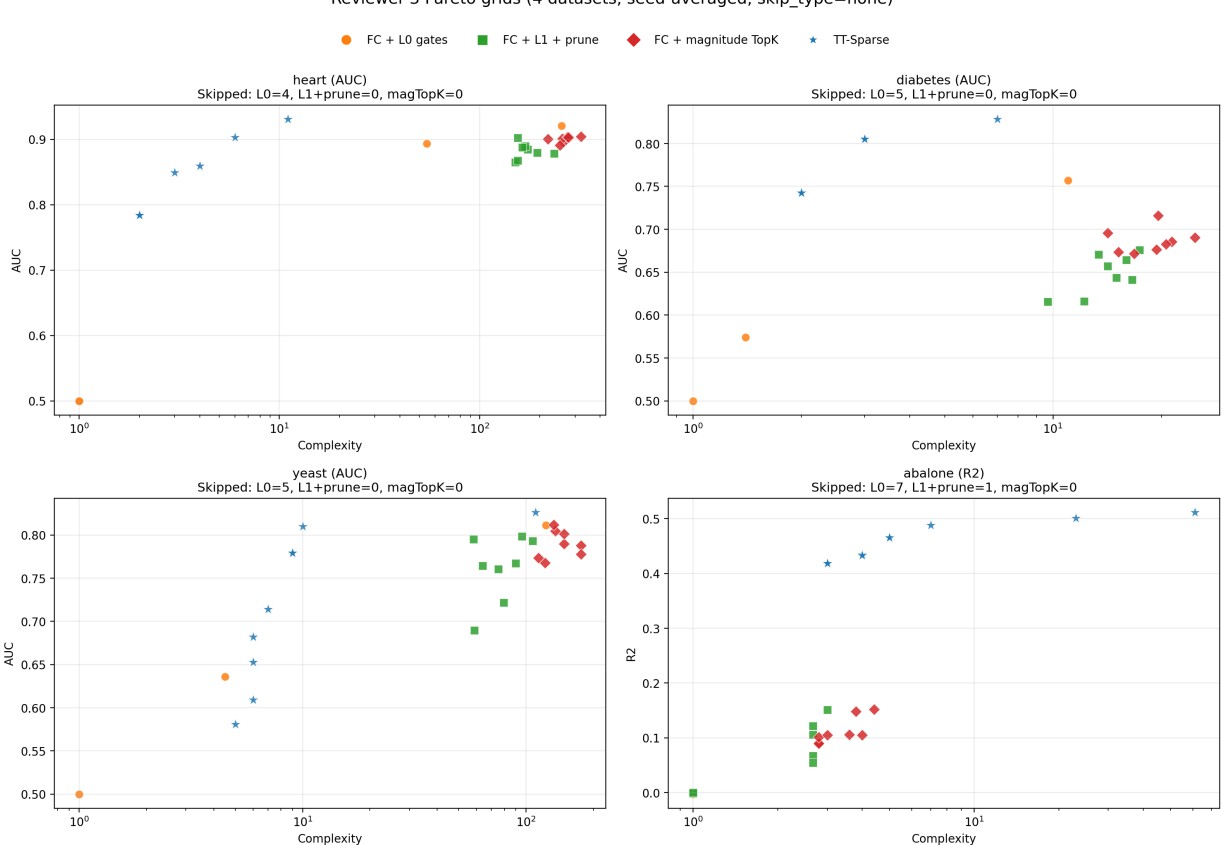

*Figure 17.* Performance–complexity Pareto grids for the TopK necessity ablation across 4 datasets and 3 task types. TT-Sparse (green) dominates the upper-left frontier on all datasets.

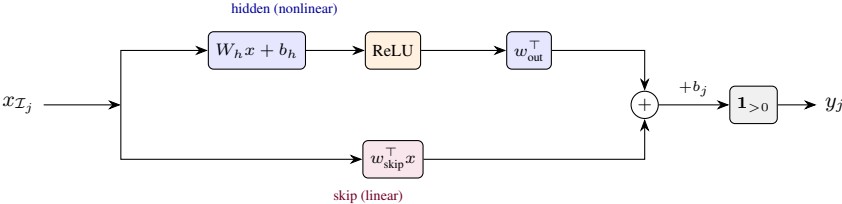

*Figure 18.* Deep LTT node. The hidden path provides universal Boolean expressivity via a ReLU layer; the skip path preserves the standard linear threshold as a residual. Both share the same TopK mask.

*Table 8.* Standard LTT vs. Deep LTT aggregate comparison. $\Delta$ = deep − standard (negative favors standard). W/L = wins/losses for deep LTT by metric.

| Task | Datasets | Mean $\Delta$ metric | W/L | Mean $\Delta$ complexity |
|---|---|---|---|---|
| Binary | 14 | $-0.013$ AUC | 1/13 | $+43$ |
| Multiclass | 6 | $-0.002$ AUC | 1/5 | $+21$ |
| Regression | 7 | $-0.024\ R^2$ | 1/6 | $-37$ |

