# OpenReview forum: "TT-Sparse: Learning Sparse Rule Models with Differentiable Truth Tables"
_ICML.cc/2026/Conference — ICML 2026 regular_

### Official Review · Reviewer_ZaQd · 2026-02-28

**Soundness:** 3
**Presentation:** 3
**Significance:** 2
**Originality:** 2
**Overall Recommendation:** 4
**Confidence:** 3

**Summary:**

This paper proposes TT-SPARSE, a framework for learning interpretable rule models on tabular data using trainable sparse “truth-table nodes” (LTT). During training, the method learns sparse connections and thresholded gating to produce binary node activations, followed by post-hoc magnitude pruning and fine-tuning. For rule extraction, it enumerates each node’s truth table and applies QMC minimization to compile the node behavior into compact CNF/DNF rules, yielding a final model in the form of “logical expressions + linear weights.” The paper reports performance–complexity trade-offs on binary classification, multiclass classification, and regression tasks, and provides readable rule examples.

**Compliance With Llm Reviewing Policy:**

Affirmed.

**Final Justification:**

The supplementary experiment significantly alleviated my concerns; I will now view it positively, but I believe the article's originality is moderate. Overall, I will raise my score to 4 points.

**Key Questions For Authors:**

See Weakness

**Limitations:**

Yes

**Strengths And Weaknesses:**

Strength

1. The overall pipeline is complete and coherent: from a trainable rule layer to explicit global rules via truth-table enumeration and logic minimization, producing human-readable rule systems.

2. The evaluation covers binary/multiclass/regression and reports results from a performance–complexity perspective; the qualitative rule examples are helpful.

Weakness

1. The paper positions Soft Top-k (hard-forward/soft-backward) as a key contribution, but it does not provide direct evidence that an explicit Top-k constraint/module is necessary. The current ablation mainly compares against a slot-based softmax selection mechanism, which is essentially another way to “select k inputs,” and does not address a more fundamental question: if we remove the explicit Top-k structural constraint and instead use more standard sparsification routes (e.g., learnable gates with L0/L1 regularization, or sparsity regularization plus magnitude pruning/fine-tuning, and then finalize the structure before extraction to control each node’s effective fan-in), can we obtain a similar Pareto frontier under the same rule-complexity budget? This comparison would materially affect my view of Soft Top-k as a main novelty/essential component.

2. For contribution 3）in line 108: QMC/Boolean minimization is a fairly standard and widely used compilation step in related work，why it can be regarded as one contribution？

---

> ### Author Rebuttal · Authors · 2026-03-31
>
> We thank R**ZaQd** for the constructive review and for highlighting our empirical evaluations. We address your two main questions below.
>
> 1. Is the explicit TopK fan-in constraint necessary, beyond standard sparsification?
>
> We appreciate this question and agree it is important, especially since our pipeline also applies post-training pruning. To address it directly, we added the exact types of baselines you suggested, plus an additional ablation to isolate the role of our separate connection-learning mechanism.
>
> We compare TT-Sparse against three alternatives that share the same backbone architecture (LTT layer with STE binarization, followed by the same classifier head) and differ only in how sparsity / fan-in are imposed before rule extraction:
>
> 1. **FC + L0 gates**: fully connected LTT with hard-concrete gates (Louizos et al., ICLR 2018), trained with an L0 penalty on expected fan-in; gates are then thresholded and the surviving structure is fine-tuned under a fixed mask.
> 2. **FC + L1 + prune**: fully connected LTT trained with L1 regularization, followed by iterative magnitude pruning and fine-tuning.
> 3. **FC + magnitude TopK**: fully connected LTT first trained without structural selection, then the top-k inputs per node are chosen by weight magnitude, followed by L1-regularized fine-tuning within the top-k mask, iterative magnitude pruning to the target fan-in, and a final fine-tuning pass.
>
> All variants are evaluated over $\lambda \in [0, 0.001, 0.005, 0.01, 0.05, 0.1, 1, 10]$, averaged across 5 seeds, and converted to DNF rules via the same QMC extraction pipeline to produce comparable performance–complexity Pareto curves.
>
> In this table we report the performance and complexity averaged over 5 seeds of the Pareto points of the models for 4 datasets.
>
> | Dataset | Metric | TT-Sparse | L0 Gates | L1+Prune | Mag TopK |
> |:--|:--|:--|:--|:--|:--|
> | **Heart** | AUC | **0.931**±0.01 | 0.920±0.015 | 0.902±0.027 | 0.904±0.029 |
> | | Cmplx. | **11**±15 | 257±85 | 155±39 | 321±174 |
> | **Diabetes** | AUC | **0.821**±0.02 | 0.757±0.010 | 0.676±0.049 | 0.716±0.057 |
> | | Cmplx. | 14±9 | **11**±4 | 17±14 | 20±7 |
> | **Yeast** | AUC | **0.846**±0.02 | 0.811±0.023 | 0.798±0.063 | 0.812±0.026 |
> | | Cmplx. | **77**±14 | 123±38 | 96±79 | 133±68 |
> | **Abalone** | R² | **0.533**±0.01 | NA** | 0.151±0.078 | 0.152±0.088 |
> | | Cmplx. | 43±16 | NA | **3**±0 | 4±3 |
> **\*\*** L0 gates failed to reduce fan-in below 18 on Abalone even at $\lambda=1.0$, making truth table enumeration intractable ($2^{18}$ entries per node) and QMC conversion infeasible within practical time and memory limits. Only at $\lambda=10$ did fan-in collapse, but at that point the model had degenerated to a trivial predictor ($R^2 \approx 0$).
>
> These results support two points.
> First, **generic sparsification alone is not sufficient** to recover the same tradeoff as TT-Sparse. Unconstrained L0/L1 approaches can still leave some nodes with large or highly uneven fan-in, which hurts the final extracted rule complexity even after pruning. Moreover, post-hoc pruning fundamentally damages learned representations: during unconstrained training, the network distributes information across all available connections, learning very high-order rules for the nodes, so when these connections are pruned afterwards, it results to huge performance drops. This is reflected in the performance gaps to TT-Sparse in the table. In contrast, TT-Sparse forces the network to learn compact representations from the start by constraining each node to k inputs throughout training.
> Second, the comparison with **magnitude TopK** shows that it is not enough to simply choose the k largest logic weights: learning connectivity through a separate mapping mechanism $W_{map}$ improves the resulting performance–complexity tradeoff.
>
> We have pushed the code producing these results to the same anonymous [repository](https://anonymous.4open.science/r/sparse-CF97) of the paper, specifically at `ablation/topk_variants_ablation.py` and provide the full Pareto grids for all 4 datasets in this figshare [link](https://figshare.com/s/468dd43508ae2af3625e).
>
> 2. Thank you for raising this point. We agree that the QMC algorithm is not novel and it has been used in the literature (to the best of our knowledge only by TTnet by Benamira et al.). Our contribution lies in the end-to-end pipeline integration, the architecture is designed such that their behavior can be completely extracted with QMC. To avoid ambiguity, we will revise the paper by merge points 2 and 3 to a single point for clarity.
>
> We hope the evaluations on the standard sparsification methods with the additional baseline clarifies our design choice and our contribution claims are clarified. Please let us know if any feedback remain to strengthen our paper.

---

> > ### Author Rebuttal · Reviewer_ZaQd · 2026-04-02
> >
> > The supplementary experiment significantly alleviated my concerns; I will now view it positively, but I believe the article's originality is moderate. Overall, I will raise my score to 4 points.

---

> > > ### Author Response · Authors · 2026-04-04
> > >
> > > Thank you for acknowledging our additional data supporting the necessity of our top-k property and the positive reassessment of our work. We believe that the core originality lies in introducing novel differentiable TopK mathematical operator with the necessary properties over its predecessors (rebuttal **BZBL** point 4.2). We identified an effective method to enable general Boolean logic learning through truth table nodes that maximizes expressivity while allowing us to control the complexity by controlling its fan-in. However, this was bottlenecked by the gradient variance and computational overhead of the existing differentiable TopK operators which is required for gradient backpropagation to learn the effective connections for our truth table inputs. Our theoretical contribution in introducing this zero-variance, lightweight soft TopK mathematical operator allowing backpropagation for our purpose of truth table learning is also reflected in the strong empirical results shown. We hope this helps clarify why we see our contribution as more than moderate in originality.

---

### Official Review · Reviewer_1Ubt · 2026-03-12

**Soundness:** 2
**Presentation:** 2
**Significance:** 3
**Originality:** 3
**Overall Recommendation:** 4
**Confidence:** 3

**Summary:**

The paper deals with the task of learning rule sets that both achieve high predictive accuracy and have low complexity. To do so, the authors introduce a new neural building block that they call 'TT-Sparse'. It (a) can be trained end-to-end in a differentiable way and (b) allows reading out sparse rules after training. The goal is to achieve global and exact interpretability, i.e., the rules describe the entire model behavior (hence 'global') without approximation (hence 'exact'). This new architecture is evaluated on 28 tabular data tasks (binary, multiclass, and regression).

**Compliance With Llm Reviewing Policy:**

Affirmed.

**Final Justification:**

I thank the authors very much for their rebuttal and their reply to my acknowledgment. This addressed most of my concerns, so I increase my overall recommendation.

**Key Questions For Authors:**

1. See the question about sparseness guarantees in weakness 2.
2. See the question about a potential characterization of the Soft TopK operator in weakness 4.
3. To double-check, what is the expressive power of the TT-Sparse model: can it express any Boolean function? That is, for any $k$ and for any Boolean function $f$ of $k$ variables, is there some TT-Sparse model $M$ whose extracted rule is $f$? How quickly do the number of parameters of $M$ grow with $k$? Would they grow differently if the depth (i.e., number of TT-Sparse blocks) would be increased rather than the width of one block?

**Limitations:**

Yes, the authors do discuss limitations. Additionally, as described in weakness 2 above, claims about achieving interpretability should be caveated if sparseness of the rule set cannot be guaranteed; and as described in weakness 3 above, claims about architectural flexibility should be experimentally supported.

**Strengths And Weaknesses:**

The paper contributes to a challenging problem at the heart of neuro-symbolic AI. It is particularly difficult to obtain the combination of global and exact interpretability via sparse rules while still achieving high accuracy. In this light, it is impressive that the authors achieve a "a new Pareto frontier in the performance-complexity landscape while achieving predictive accuracy competitive with SOTA non-pretrained tabular model, TabM" (l. 116-9).

Despite the strengths, the paper also has some weaknesses:

1. *Related work*: The paper discusses several approaches to global and exact interpretability in section 2. However, I was surprised that the field of inductive logic programming is not mentioned. This approach, too, aims to generate a finite rule set from training data. For a recent overview, see

   * Cropper, Andrew, et al. "Inductive logic programming at 30." Machine Learning 111.1 (2022): 147-172.

   To name but one interesting model that could serve as a baseline within this family of approaches:

   * Evans, Richard, and Edward Grefenstette. "Learning explanatory rules from noisy data." Journal of Artificial Intelligence Research 61 (2018): 1-64.

2. *Interpretability*: As the authors note on p. 1, just providing a global and exact rule set for a model is not enough to achieve the benefits that one would want from interpretability (e.g., accountability, auditability, human-understandability, etc.) because the rule set could be so large that it is as complex/opaque as the parameters of the model itself. In the example presented in figure 2, the TT-Sparse model does achieve a low rule complexity. But are there any general guarantees (in the form of a mathematical theorem) that ensure that the rule set extracted from the trained model is sparse? It seems difficult to provide such guarantees, because the TT-Sparse model employs the Quine-McCluskey algorithm to minimize the found rules, and this algorithm computes an NP-hard problem. However, without such a guarantee, claims about achieving interpretability should be caveated.

3. *Architectural flexibility*: The authors note that "TT-Sparse offers significant architectural flexibility; it can be seamlessly integrated into standard neural pipelines" (l. 393-7). Unless I have missed something, the authors do not actually emperically test this flexibility. Without such an empirical example it is difficult to assess the flexibility claim. Moreover, if I understand correctly, the trained models in the paper use just one TT-Sparse block; would a deep version of the model with several TT-Sparse blocks achieve yet better results?

4. *The Soft TopK operator*: The main technical contribution of the paper is to introduce the Soft TopK operator. The challenge it has to solve is—intuitively speaking—to find the k-many, most important input features for a node, so they can be later represented by a Boolean function of just k-many arguments. A priori, there are many ways to mathematically formalize this intuitive idea (e.g., (global) Shapley values have a similar aim). Is there an argument why to choose the presented definition and not another one? For example, can the properties that uniquely characterize binary entropy $H$ be used to give some kind of uniqueness result for the definition of the Soft TopK operator? Otherwise, this leaves the reader wondering if other definitions could not further improve the results.

5. *Math presentation*: There are several mathematical imprecisions that make the paper difficult to read. For example:

   * Section 3.1 starts by defining a 'space' of operators. Though later, in line 213, one learns that actually just one operator, $S_k$ (relative to a choice of $k$) is defined.
   * The notation $\mathcal{P}_k$ is not defined (though, one assumes that it is the set of vectors in $[0,1]^n$ whose components sum to $k$). Also, $\sigma$ is not defined (one assumes it is the sigmoid function).
   * The index of the sum in line 188 should be $i$ and not $j$.
   * The reference to '3.1' in line 214 seems to be wrong.
   * The 'stop_gradient' function in line 243 is not defined.

6. *Terminology*: There are several expressions that were not quite clear to me:
   * 'learn effective truth table logic' (l. 192). Does this mean 'learning a Boolean function'? Is this the same as 'learnable Boolean logic' (l. 223)?
   * What does it mean exactly that the TT-Sparse block 'integrates high-order feature interaction logic through the TT-Sparse layer with linear single-order features' (l. 247-250)? Specifically, what is a 'high-order feature interaction logic'?

---

> ### Author Rebuttal · Authors · 2026-03-31
>
> We thank R**1Ubt** for the constructive feedback. We address your main points below.
> 1. We agree that Inductive Logic Programming (ILP) is a relevant family of approaches that we should have discussed. We will add a discussion of ILP to our Related Work section, specifically to the Neuro-Symbolic Approaches subsection. To empirically ground this discussion, we implemented the Differentiable ILP (DILP) model by Evans et al. from the DILP-Core library as a baseline and evaluated it on the binary classification datasets using our evaluation protocol. The comparison is limited to binary classification because of its architectural limitation, it's designed to learn rules for 1 predicate. The results can be found in this figshare [link](https://figshare.com/s/582a1bde20a5b6ac253a).
>
> DILP occupies a different region of the performance-complexity landscape with simpler rule sets and lower performance. We include this comparison in the revised paper's experimental section.
>
> 2. You are correct that because QMC solves an NP-hard minimization problem, there is no strict a priori mathematical guarantee on the maximum sparsity of the final rule set, as it inherently depends on the target data distribution. However, our architecture strongly bounds the empirical complexity by design: L1-pruning removes redundant connections to LTT nodes, and the Soft TopK operator strictly bounds the maximum fan-in ($k$) for each node on only k binary variables. Any such function admits an exact canonical Boolean representation of size bounded solely by k. QMC then guarantees the shortest possible exact representation of that trained logic.
>
> 3. We have made the claim of architectural flexibility as we are able to plug TT-Sparse into any of the 3 tabular tasks by modifying the classifier layer, which is a limitation of the existing rule models. We understand that "seamlessly integrated into standard neural pipelines" is overstated without empirical testing into other kind of models and thus we revise this statement. Regarding deep LTT, see point 7. Thank you for the feedback.
>
> 4. You are right that generally the challenge is to find the k most important input features to the node, which is the goal of the $W_{map}$ module with the Soft TopK backpropagation mechanism. Regarding SHAP, they serve fundamentally different functions as they provide post-hoc, global feature attribution by calculating marginal contributions across $2^n$ combinations, not differentiable operators that can be applied to route gradients for backpropagation. Why choose the binary entropy $H$? Because it acts as a barrier function to ensure the weightage of the gradients propagated remain in the open interval (0,1) based on the magnitudes, never collapsing to 0 to allow the network to continuously explore the feature space. Given $H$, our soft topk is then uniquely derived (l185) via Lagrange multipliers (Appendix C), which improves the previous differentiable topk operators on stability and throughput (see rebuttal 1 point 4.2).
>
> 5. We thank you for pointing out the math presentation details. We will correct them accordingly.
>
> 6. You're right, we use "Boolean logic" and "truth table logic" interchangeably. High-order feature interaction logic refers to rules that combine multiple distinct features, as opposed to single-order features.
>
> 7. Addressing key question 3 regarding LTT nodes expressivity: A single LTT node with k inputs is formally a linear thresholded function, so it cannot approximate the entire space of $2^{2^k}$ Boolean functions $\{0,1\}^k\to\{0,1\}$ of that node's input. On the other hand, the TT-Sparse model is structurally a depth-2 threshold circuit, so each LTT node can isolate a specific $k$-dimensional minterm, by aggregating the node's outputs, it can express any binary function $f:\{0,1\}^n\to\{0,1\}$ of the whole input space.
>
> This insight drove us to propose a variant of the LTT node to not only allow the whole TT-Sparse model express any function of the whole input space, but each LTT node to express any function of the node's inputs. In this variant, each LTT node contains a hidden layer of size $2^k$ with ReLU activations for nonlinearity, guaranteeing the node to reach every vertex in the Boolean hypercube (Baum, 1988, On the Capabilities of Multilayer Perceptrons) and still enables training through gradient backpropagation. We perform synthetic experiments to evaluate the capabilities of the LTT nodes to learn random truth table logic of different bit sizes, and find that the deep LTT variant can exactly fit the functions where the original variants fails to, verifying our theoretical grounding. We also empirically evaluate its performance on the real datasets by applying it to the TT-Sparse model, and find that the theoretical expressivity does not directly translate to real-world gains in performance on real datasets. The full experimental results can be found in the shared figshare link.

---

> > ### Author Rebuttal · Reviewer_1Ubt · 2026-04-03
> >
> > I thank the authors very much for the detailed rebuttal, in particular the relation to DILP and the deep LTT! Most of my questions are answered, though the following remain.
> >
> > * The main one is regarding the authors' comment 2: In sum, there is (at least currently) *no* mathematical guarantee on the sparsity of the final rule set—neither a deterministic one nor a statistical one (depending on the target distribution). The authors also note in their comment that "QMC then guarantees the shortest possible exact representation of that trained logic"; however, as already discussed, that guarantee seems practically irrelevant, since the problem that the Quine-McCluskey algorithm computes is NP hard (at least absent a parametrized complexity result that establishes feasibility within the parameters relevant to the model). Thus, as noted in my review, this situation seems to be at odds with the motivation of the paper: The motivation of the paper seems to be to achieve "human-understandable" interpretability by sparse rule sets, but presently no sparsity guarantees can be given. I think it is important to be transparent about this in the paper. Unless I am missing something, it would be good to at least include a careful—quantitative and qualitative—analysis of how human-understandable the obtained rule sets are in the experiments.
> >
> > * I also have a smaller remark regarding the authors' comment 6 that "High-order feature interaction logic refers to rules that combine multiple distinct features, as opposed to single-order features". This terminology is a bit confusing, since this usage seems to be different from the sense of "higher-order" in logic. For example, second-order logic is characterized by the fact that, in the language, one can not only quantify over objects (as in first-order logic) but also over sets of objects. But a formula involving many relation symbols/features and, e.g., implications between them, would still be first-order. I recommend the authors to adjust or at least explain their usage of the term "higher-order" in the paper.

---

> > > ### Author Response · Authors · 2026-04-06
> > >
> > > We thank R**1Ubt** for the follow-up and acknowledging our results.
> > >
> > > 1. We appreciate this concern and agree that human-understandability is essential. We clarify 2 distinct points: (a) The computational complexity of QMC, and (b) the empirical sparsity of the rules.
> > >
> > > (a) To briefly set up the terminology: QMC minimizes a Boolean function's truth table into a compact DNF (OR-of-ANDs). Each AND-clause is an *implicant* and each variable in it a *literal* (e.g., $(A \land \neg B) \lor C$ has 2 implicants, 3 literals). The *density* is the fraction of inputs mapping to 1; higher density means harder minimization.
> > >
> > > You are correct that finding a minimum-cost cover of prime implicants is NP-hard in general. However, TT-Sparse constraints the fan-ins of each node to $k$ binary inputs ($k \in \{4,5,6\}$) before further pruning. The NP-hardness is w.r.t the number of inputs, and our architecture bounds this. We benchmark QMC runtime across fan-in values from 2 to 9 (avg. over 5 random truth tables):
> > >
> > > |$k$|rows|d=0.25|d=0.50|d=0.75|
> > > |:-:|:-:|:-:|:-:|:-:|
> > > |2|4|0.05ms|0.05ms|0.14ms|
> > > |3|8|0.06|0.15|0.18|
> > > |4|16|0.14|0.50|0.50|
> > > |5|32|0.51|1.48|2.65|
> > > |**6**|**64**|**1.77**|**5.68**|**9.11**|
> > > |7|128|7.50|24.9|40.8|
> > > |8|256|30.8|92.5|157|
> > > |9|512|101|346|900|
> > >
> > > At our maximum fan-in $k=6$, the QMC completes in under 10 ms per node, which is negligible, so intractability does not apply to our setting. Beyond computational benefits, referring to our ablation in rebuttal to R3 (ZaQd) point, the architectural fan-in top-k constraint produces better rule models: TT-Sparse consistently outperforms generic sparsification baselines (L0 gates, L1 regularization, magnitude TopK) because forcing the network to learn compact $k$-input representations from the start yields better rules than with unconstrained connections.
> > >
> > > (b) You are correct that there is no closed-form worst-case sparsity bound for arbitrary data distributions, as with existing state-of-the-art interpretable models we position ourselves against such as GOSDT (ICML 2020), GLRM (ICML 2019), and other recent neuro-symbolic methods. Similarly, as in the literature, rule complexity is controlled via architectural constraints and regularization which is empirically evaluated with the Pareto grid of performance-complexity.
> > >
> > > We agree that human-understandability is not automatic and following our reference of Lage et al. (2019) in our Introduction, rules with complexity above 50 are hard to understand.
> > >
> > > The worst-case complexity through TT-Sparse's architectural constraints is $Mk2^k$ with $M$ nodes and $k$ binary inputs per node, **before** pruning, QMC minimization, and don't-care simplification. Each node has a truth table of at most $2^k$ rows, so each node's worst-case complexity is $k2^k$ by listing all minterms with $2^k$ conjunctions with $k$ literals each. In practice, learned functions are far simpler.
> > >
> > > We conducted a 2-part complexity analysis across 5 datasets (creditg, eye_movements, iris, boston, ecoli) and 5 seeds:
> > >
> > > First, we extract and minimize every non-trivial LTT node's Boolean function across all trained models and measure the resulting implicant and literal counts:
> > >
> > > |$k$|Implicants|Literals|Density|
> > > |:-:|:-:|:-:|:-:|
> > > |2|1.20|1.69|0.43|
> > > |3|1.59|2.91|0.39|
> > > |4|2.35|5.43|0.37|
> > > |5|3.52|9.77|0.35|
> > > |6|5.32|17.1|0.34|
> > >
> > > The average contribution to the overall complexity (literals) of each node remains moderate even at $k=6$, with mean density consistently below 0.43 and decreasing with $k$. This means the trained LTT nodes converge to sparse Boolean functions well below the worst-case bound. To see how the complexity contributed by these nodes fit into the overall architecture to produce the final rule set, we run an experiment to see how the performance, complexity, mean fan-in, # of rules changes as we vary the pruning strength ([full data](https://figshare.com/s/e4b7f0c0338432dd5340))
> > >
> > > |Dataset|Prune%|Metric|Cmplx|Fan-in|Rules|Baseline|
> > > |:-|:-:|:-:|:-:|:-:|:-:|:-|
> > > |creditg|0|.773|515|6.0|50|GLRM: .685, c=56|
> > > ||90|.754|**49**|1.7|12||
> > > |eye_mov.|0|.620|294|5.0|50|GLRM: .628, c=16|
> > > ||70|.606|59|2.2|31||
> > > ||90|.596|**23**|1.3|17||
> > > |iris|0|1.00|45|5.0|20|NeuRules: .988, c=25|
> > > ||70|1.00|**16**|2.0|14||
> > > |boston|1S0|.793|114|4.6|36|GLRM: .654, c=74|
> > > ||70|.771|**44**|2.2|29||
> > > |ecoli|0|.982|190|6.0|50|NeuRules: .934, c=201|
> > > ||70|.975|**46**|2.1|43||
> > >
> > > With the analysis of the complexity produced by each LTT node and the # of nodes we can control through hyperparameter and pruning strength, we demonstrate how TT-Sparse can produce rule sets below complexity 50 while retaining superior performance to interpretable baselines. We will add these quantitative analyses and clarify this point of human-understandability as an empirical objective.
> > >
> > > 2. We also thank you for the terminology remark. We agree that “higher-order” is potentially misleading in the logic sense you mention, and in the revision we will replace it with clearer wording such as multi-feature interaction rules.

---

### Official Review · Reviewer_BZBL · 2026-03-12

**Soundness:** 3
**Presentation:** 3
**Significance:** 2
**Originality:** 2
**Overall Recommendation:** 2
**Confidence:** 4

**Summary:**

This paper proposes the TT-SPARSE layer, a versatile neural building block for learning a classification model based on the learned sparse and interpretable rules. Specifically, the method introduces Learnable Truth Table (LTT) nodes, which learn Boolean logic over a small subset of input features while maintaining sparsity. A key technical component is a soft Top-k operator that enables differentiable training while enforcing discrete feature selection in the forward pass. After training, the learned nodes can be converted exactly into compact Boolean rules, providing global and exact interpretability. Experiments on tabular datasets show that TT-SPARSE achieves competitive predictive performance while producing rule sets with lower complexity compared to existing interpretable baselines.

**Compliance With Llm Reviewing Policy:**

Affirmed.

**Final Justification:**

The rebuttal addresses some of my concerns. Please refer to the details in my Rebuttal Acknowledgement.

**Key Questions For Authors:**

1. Motivation.

I do not fully follow the discussion of the limitations of existing neuro-symbolic methods in lines 85--89 and 97--101. Could the authors further clarify what is meant by “impose restrictive logical forms that limit expressivity” and “approximate discrete logic through continuous functions”? As written, these statements do not seem fully precise to me. For example, one can often interpret the output of a neuro-symbolic method as a weighted OR over the AND of all symbol instantiations, such as $0.7(c_1 \wedge c_2) \vee 0.1(\bar{c}_1 \wedge c_2) \vee 0.1(c_1 \wedge \bar{c}_2) \vee 0.1(\bar{c}_1 \wedge \bar{c}_2)$, where the weights are provided by the neural component. In this sense, I would appreciate a more precise explanation of the claimed limitations.

2. Questions on Section 3.1.

2.1 It seems that the proposed soft top-k operator is essentially a projection onto the simplex. Is this interpretation correct?

2.2 I am also unsure about the role of the entropic regularizer in line 161. Intuitively, increasing entropy would seem to make the node output less interpretable, which appears contrary to the paper’s goal. Could the authors clarify this design choice?

2.3 It seems that the paper never explicitly defines the loss function $\mathcal{L}$. Could the authors provide this definition clearly?

2.4 It would also be helpful if the authors could clarify the main technical challenge addressed by the proposed method. At present, the core contribution appears to be mainly the simplex projection step, so a more explicit explanation of the methodological difficulty would strengthen the presentation.


3. Questions on Sections 3.2 and 3.3.

3.1 From Section 3.2, it appears that the final prediction depends on both $x$ and $z$. If so, the prediction may not be fully interpretable, since it still relies on the raw input $x$. Could the authors clarify this point? In particular, why not use $x$directly if it remains part of the prediction process?

3.2 In Section 3.3, the formulation in line 235 does not seem dimensionally consistent. Earlier, $z_j()$ appears to be applied to an $n$-dimensional vector such as $x$, whereas here it is applied to $v_i$​, which is a $k$-dimensional vector. Could the authors clarify this formulation?

3.3 One motivation of the paper is that *“when possible, one should prefer inherently interpretable models over explanations of opaque ones, where transparency is a property of the model itself”*. However, based on the rule extraction phase, the rules seem to be obtained in a **post hoc** manner. My impression is that the model does not actually rely on the extracted rules for prediction, but instead predicts through $f_{\mathrm{cls}}$ defined in line 256. Could the authors clarify whether the extracted rules are intrinsic to the prediction process or only used for explanation afterward?


4. Experiments.

4.1 Why do the experiments focus only on tabular datasets? According to line 123, the comparable methods have also shown effectiveness on image datasets such as ImageNet. Evaluating on a broader range of data types would make the empirical study more convincing.

4.2 My understanding is that the main novelty of the paper lies in the proposed soft top-k operator. If so, it would be important to compare it directly against related differentiable top-k methods discussed in lines 114--121. However, I could not find experimental results against those baselines. Could the authors comment on this omission?

**Limitations:**

yes

**Strengths And Weaknesses:**

**Strengths.**

The paper addresses the difficulty of discrete top-k selection by introducing a soft top-k operator, which enables gradient-based optimization.

The method can also produce sparse rule-based explanations for the final prediction, which improves interpretability.

**Weaknesses.**

The level of novelty and the main technical challenge are still unclear to me. Please see the detailed comments in the Key Questions section.

In addition, comparisons with the most relevant baselines appear to be missing, especially methods that also rely on differentiable or soft top-k operators.

---

> ### Author Rebuttal · Authors · 2026-03-31
>
> We thank R**BZBL** for the detailed review and for recognizing the interpretability benefits of our approach. We address the constructive feedbacks below:
>
> 1. We completely agree that a weighted combination of rules, such as $0.7(c_1 \wedge c_2) \vee 0.1(\bar{c}_1 \wedge c_2)$, is highly interpretable. By "restrictive logical forms," we refer to strict architectural topologies enforced by existing methods. For instance, RRL (Wang et al., NIPS 2021) relies on fixed alternating layers of soft-AND and soft-OR operators, which prevents flexible logical mixing, lacks negation, and struggles with long rules due to combinatorial explosion. NeuRules (Xu et al., NIPS 2025) is limited to sequential AND-only rule lists with fixed lengths and single-feature thresholds (e.g., unable to learn Age > 30 AND Income > 50k). DiffLogicNet (Petersen et al., NIPS 2022) learns truth tables but is constrained by a fixed 2-arity per node and random initialization. These formulations enables end-to-end gradient-based optimization, however, are limited in the binary functions that can be learned. TT-Sparse bypasses these constraints by using LTT nodes to learn Boolean functions over $k$ inputs, maximizing local expressivity. We further discuss our improvements over DiffLogicNet's efficiency in Sec 3.3.
>
> 2.1. Yes, our Soft TopK is indeed a projection onto a k-capped simplex, solving the entropy optimization objective with Lagrangian multipliers
>
> 2.2. The entropic regularizer $\tau$ is only used during training to ensure the routing weights receive smooth gradients, so the connections can change during training to find meaningful connections. The final model for rule extraction uses the discrete TopK so the entropic regularizer does not compromise interpretability, this is highlighted explicitly in Sec 3.1 LTT nodes.
>
> 2.3 Thank you for pointing out this presentation detail. We use standard cross-entropy for classification and MSE for regression.
>
> 2.4 Truth tables are maximally expressive but scale as $2^k$, so each node must select a small k-subset from n features. This is a discrete combinatorial routing problem inside a gradient-based pipeline. The core difficulty is making this selection differentiable without introducing gradient variance (Perturbed) or massive overhead (Sinkhorn) which our Soft TopK achieves (see point 4.2)
>
> 3.1 The classifier layer is linear, so each weight directly corresponds to either a single feature (from x) or a higher-order Boolean rule (from z). The extracted rule set includes both terms all with exact weights. Removing x degrades performance (Ablation H) because single-feature effects would need to be redundantly encoded as trivial 1-input truth tables
>
> 3.2 Thank you for pointing out this mistake, we will correct our notations accordingly
>
> 3.3 The rules are **fully intrinsic**, exact transparency is a fundamental property and goal of this model. The rules and the neural form are functionally equivalent and produce identical predictions for all inputs. The predictive performance and complexity we report empirically represent the rule inferences, which is identical to the model's inferences via weights.
>
> 4.1 We appreciate the reviewer pointing out the task which we apply this. We benchmark TT-Sparse against global and exact rule models in the tabular tasks domain, NeuRules, RRL, GLRM (ICML 2019), GOSDT (ICML 2020), etc, which is fundamentally different from interpretability for vision (e.g. concept bottlenecks or saliency maps). The paper mentioned in l123 proposes a novel topk-based loss for image classification, not an interpretable method. We agree that exploring our interpretability application for vision tasks is a future direction that is worth exploring.
>
> 4.2 Thank you for pointing this out, Soft TopK is a core novelty of our paper, because we see that the existing differentiable TopK has fundamental limitations (Sec 2) for our application of gradient backpropagation for the connection learning mechanism. Xie et al. (NIPS 2020) uses an iterative Sinkhorn algorithm to solve an optimal transport problem, while Cordonnier et al. (CVPR 2021) perturbs score with Gaussian noise to yield soft outputs. We conduct an empirical analysis to compare our Sigmoid-based TopK in terms of computational overhead and output variance.Full operator comparison (mean across benchmark cases, GPU CUDA, float32, batch=64)
>
> | Operator | Forward (ms) | Backward (ms)  | Mem (MB) | Throughput (samples/s) |
> |-|-|-|-|-|
> | SoftTopK (ours) | 4.08 | 1.25 | 3.4 | 36,177 |
> | Perturbed | 3.46 | 1.15  | 247.4 | 31,939 |
> | Sinkhorn | 41.0 | 32.2 | 1,075.7 | 1,980 |
>
> | Metric | SoftTopK & Sinkhorn | Pert. (32 samples) | Pert. (256 samples) |
> | :- | :- | :- | :- |
> | Grad. Var. (Mean) | 0.0 | 200.29 | 22.10 |
> | Loss Std. | 0.0 | 1.80 | 0.39 |
>
> Because Sigmoid is a deterministic soft approximation, it has zero gradient variance. Our efficiency and stability are deliberate design choices that enable the scalability we demonstrate in TT-SPARSE.

---

> > ### Author Rebuttal · Reviewer_BZBL · 2026-04-03
> >
> > R1, R2.1, R4.1, and R4.2 address my concerns.
> >
> > 1. Thanks for your explanations in R2.2 and R2.3. It would be appreciated if you could add these clarifications on the role of the entropic regularizer and the choice of $\mathcal{L}$ in the paper.
> >
> > 2. From R2.4, I understand the technical challenges. But since the proposed method is essentially a projection on the simplex, I am not sure whether the contributions are sufficient. I need to discuss with other reviewers.
> >
> > 3. In R3.1, the authors state that removing $x$ degrades performance. And this actually corresponds to my concern in Q3.1. It seems the method has to rely on the raw input $x$; in this sense, the prediction is not fully interpretable.
> >
> > 4. The authors mention they will change the incorrect formulation in line 235. I think this incorrect formulation prevents readers from understanding the core contributions (i.e., rule extraction) of the paper. However, they don't give the correct formulation in the rebuttal, which still makes it hard for me to understand the rule extraction step.
> >
> > 5. R3.3 hasn't addressed my concerns. I'd like to paraphrase my question here: Are rules extracted in a post-hoc manner? If not (then it's intrinsic), then why are the rules and the neural form functionally equivalent?

---

> > > ### Author Response · Authors · 2026-04-05
> > >
> > > We thank R**BZBL** for acknowledging our additional experiments in our rebuttal. Our rebuttal was condensed due to the 5000-character limit thus we hope to clarify your concerns further here.
> > >
> > > 1. We appreciate the feedback, the role of $\tau$ will be expanded upon on l209-211 and the loss function will be mentioned explicitly.
> > >
> > > 2. The appropriate simplex projection according to the entropy objective is a technical solution to our general objective of learning performant rule sets with low complexity by learning generalized Boolean/truth table logic with controllable fan-ins, which uses the simplex projection as a differentiable routing mechanism for the connections. See point 1 of our rebuttal to reviewer 3 (ZaQd) to see why this is necessary.
> > >
> > > 3. Yes, the method relies on the raw input $x$ through the skip connections, and it doesn't compromise interpretability. This is very important to understand and we hope to clarify this by providing the [visualization](https://figshare.com/s/2db22701727ce8096e21) of a simple example of a TT-Sparse model with 2 LTT nodes on a dataset with 3 input features. Say we don't prune any of the connections in the model, the final classifier layer will have 5 nodes (2 from the LTT nodes outputs and 3 from the original features) connected to the final classification node, thus producing 5 rules (2 from LTT nodes via truth table enumeration and QMC conversion and 3 directly from the skip connections), with exact weights attached to each of them contributing to the outputs, plus the bias term from the linear layer. For a real-world dataset example of how this rule set looks like, refer to our paper's Figure 2, depicting the rule set learned for Heart dataset (it's a larger dataset with more than 3 features, the rule set produced is after L1 pruning as explained in Sec 3.2). Therefore, the skip connections does not compromise exact interpretability in any way, as the produced rule set is exactly equivalent to the neural form, in fact, the skip connections enhances interpretability through the regularization it provides by providing single-feature signals (Ablation H).
> > >
> > > 4. Apologies, we couldn't provide the corrected formulation as we didn't have any more space due to the 5000-character limit. We define a localized version of $z_j$ that calculates the discrete output for $k$-dimensional inputs. Let $\mathcal{I}_j \subset \{1,\dots,n\}$ denote the selected feature indices for node $j$, and $w^{(j)} \in \mathbb{R}^k$ the corresponding restricted weight vector. For each $v_i \in \{0,1\}^k$:
> > >
> > > $$y_j(v_i) = \mathbb{I}(v_i^\top w^{(j)} + b_j > 0)$$
> > >
> > > The full truth table $\mathcal{T}_j = \{(v_i, y_j(v_i)) \mid v_i \in \{0,1\}^k\}$ enumerates all $2^k$ binary input combinations over the selected features only. The minterms $\mathcal{M}_j = \{v_i \mid y_j(v_i) = 1\}$ are then minimized via Quine-McCluskey to yield the final Boolean rule. This is corrected in the paper.
> > >
> > > 5. We refer to point 3 to help address this concern. Each rule corresponds to a node in the classifier layer that is converted through truth table enumeration and Quine-McCluskey, that is the goal of our paper. The rules are extracted at the end after the model is trained, or it can essentially be extracted anytime throughout the process, as the nodes are equivalent to a particular DNF equation through the rule extraction process that can be done anytime. To further address your point in the review, "*when possible, one should prefer inherently interpretable models over explanations of opaque ones, where transparency is a property of the model itself*" this was our goal when we mentioned this motivation based on Rudin (2019), to build models where the rules is functionally equivalent to the model, thus the empirical results reported uses the rules' inferences.
> > >
> > > We hope this clarifies the functional equivalence of our model and the rules and the technical significance of our contribution. We genuinely appreciate your thoughtful feedback through these exchanges. If our responses addresses your concerns and changes your assessment, we hope you will consider raising your score.

---

### Decision · Program_Chairs · 2026-04-30

**Decision:**

Accept (regular)

**Comment:**

Reviewers generally appreciated the interpretability and the challenges of the approach. Of main concern in this discussion was the contribution of the soft top-k operator and whether it truly provides a non-trivial delta over existing work. The rebuttal period largely addressed this concern, with the authors adding new experiments demonstrating improvements that their approach brings over generic sparsification techniques and other clarifications distinguishing it from other top-k selection techniques.

Going through the discussion, I found most of the main concerns brought up by the reviewers to be addressed. One reviewer agreed and lean to accept. For the other two reviewers, upon reading the final discussions, I found the remaining concerns to either be addressed and/or minor comments. I would encourage the authors to take this feedback into account and ensure transparency in their claims of interpretability and the practical limits of their approach. Therefore, my recommendation here is to accept.

A fair criticism that was brought up is whether the approach is useful beyond the small tabular settings, especially due to the reliance on an NP-hard algorithm. This means that, even though it has higher performance than generic sparsification techniques, it does incur a significantly large cost in compute for relatively small problems, and will struggle to scale to settings that other techniques can easily handle. Therefore, while the paper's findings can be useful for a small subset of the ICML community focused on such tabular settings, it is hard for me to argue that the findings are broadly useful to the larger community, as it seems likely to be impractical for common use cases in ML.